# Emr1 regulates the number of foci of the endoplasmic reticulum-mitochondria encounter structure complex

Faiz Rasul[1], Fan Zheng[1], Fenfen Dong[1], Jiajia He[1], Ling Liu[1], Wenyue Liu[1], Javairia Yousuf Cheema[1], Wenfan Wei [1✉] & Chuanhai Fu [1✉]

The endoplasmic reticulum-mitochondria encounter structure (ERMES) complex creates contact sites between the endoplasmic reticulum and mitochondria, playing crucial roles in interorganelle communication, mitochondrial fission, mtDNA inheritance, lipid transfer, and autophagy. The mechanism regulating the number of ERMES foci within the cell remains unclear. Here, we demonstrate that the mitochondrial membrane protein Emr1 contributes to regulating the number of ERMES foci. We show that the absence of Emr1 significantly decreases the number of ERMES foci. Moreover, we find that Emr1 interacts with the ERMES core component Mdm12 and colocalizes with Mdm12 on mitochondria. Similar to ERMES mutant cells, cells lacking Emr1 display defective mitochondrial morphology and impaired mitochondrial segregation, which can be rescued by an artificial tether capable of linking the endoplasmic reticulum and mitochondria. We further demonstrate that the cytoplasmic region of Emr1 is required for regulating the number of ERMES foci. This work thus reveals a crucial regulatory protein necessary for ERMES functions and provides mechanistic insights into understanding the dynamic regulation of endoplasmic reticulum-mitochondria communication.

[1] Ministry of Education Key Laboratory for Membrane-less Organelles & Cellular Dynamics, CAS Center for Excellence in Molecular Cell Sciences, Hefei National Laboratory for Physical Sciences at the Microscale, School of Life Sciences, Division of Life Sciences and Medicine, University of Science and Technology of China, 230027 Hefei, People's Republic of China. ✉email: weiwf@ustc.edu.cn; chuanhai@ustc.edu.cn

Membrane contact sites enable intracellular organelles to communicate with each other and thus are essential for cell function. Different membrane contact sites have been discovered and have been shown to play crucial roles in many cellular activities, including lipid transport, mitochondrial fission, and autophagy[1]. Creation of membrane contact sites requires a wide range of structural proteins who work in concert to tether the two membranes of the apposed organelles[1,2]. In addition, functional and regulatory proteins are required in a contact site-dependent manner to execute different cellular functions[2]. The list of the structural, functional, and regulatory proteins at membrane contact sites has been expanding but how these different types of proteins coordinate to function awaits further study.

In yeasts, the endoplasmic reticulum (ER)-mitochondria encounter structure (ERMES) complex plays an important role in mediating the formation of ER-mitochondria contact sites[3–5]. The ERMES complex comprises four core components (Mmm1, Mdm10, Mdm12, and Mdm34). The maintenance of mitochondrial morphology protein 1 (Mmm1) and the mitochondrial distribution and morphology protein 10 (Mdm10) are transmembrane proteins residing on the ER membrane and the mitochondrial outer membrane, respectively. Mdm12 and Mdm34 are cytosolic, functioning to bridge Mmm1 and Mdm10[5–10]. Malfunctional ERMES core components dramatically affect mitochondrial morphology, giving rise to spherical and/or grape-like mitochondria[7–9,11]. These morphological phenotypes are associated with the defective function of lipid transport mediated by the ERMES complex. Indeed, Mmm1, Mdm12, and Mdm34 are SMP (synaptotagmin-like mitochondrial-lipid binding protein) domain-containing proteins, functioning to mediate the lipid transport between the ER and mitochondria[12,13]. In addition to lipid transport, the ERMES complex regulates mitochondrial fission[14,15], mtDNA inheritance[6,16], the sorting and assembly machinery complex (SAM, required for the biogenesis of β-barrel outer membrane proteins on mitochondria)[17], and mitophagy[18]. Although many functions of the ERMES complex have been discovered, the molecular mechanisms underlying dynamic regulation of ERMES formation remain elusive.

Attempts to identify the regulatory proteins of the ERMES complex have led to the discovery of Gem1[19–21], Lam6[22], and Tom7[23,24]. However, it appears that none of these identified proteins significantly affects ERMES formation. It is, therefore, still an open question what proteins are responsible for promoting ERMES assembly and regulating the number of ERMES foci.

Here, we present a regulatory protein of the ERMES complex, i.e., Emr1 (ERMES regulator 1), in the fission yeast *Schizosaccharomyces pombe*. The absence of Emr1 significantly decreases the number of ERMES foci and Emr1 interacts and colocalizes with the ERMES core components Mdm12. Hence, Emr1 is likely a component of the ERMES complex and plays a crucial role in regulating ERMES functions.

## Results

**The absence of Emr1 leads to abnormal mitochondrial morphology.** In an attempt to identify uncharacterized genes regulating mitochondrial morphology and dynamics by microscopy-based screening, we observed dramatically reduced mitochondrial mass in a mutant strain lacking the gene *SPAC8C9.19* (Fig. 1a). According to the fission yeast (*Schizosaccharomyces pombe*) database Pombase[25], *SPAC8C9.19* encodes a small transmembrane protein of ~7 kDa and is conserved in fungi. In addition, it has an ortholog in budding yeast known as *MCO6/YJL127C-B*. Although the function of this gene family has never been characterized, the budding yeast ortholog Mco6 has been shown to

localize to mitochondria[26,27]. Owing to the role of *SPAC8C9.19* in regulating the number of ERMES foci (see below), we referred to *SPAC8C9.19* as *emr1* (ERMES regulator 1) and its protein product as Emr1.

As shown in Fig. 1a, mitochondrial mass in *emr1Δ* (*emr1*-deletion) cells was reduced dramatically. Instead of forming tubular structures, almost all mitochondria in *emr1Δ* cells displayed abnormal morphology, which was categorized as fragmented/aggregated, spherical, and others. Additionally, 19.2% of the *emr1Δ* cells (n = 344) lost mitochondria completely (Fig. 1b). These mitochondrial abnormalities were correlated with impaired cell growth on YE plates containing glucose (fermentable medium), and the poor growth of the *emr1Δ* cells was exacerbated on YE plates containing glycerol (non-fermentable medium), indicative of compromised mitochondrial function (Fig. 1c). Despite the multiple types of abnormalities observed in *emr1Δ* cells, mitochondria of *emr1Δ* cells were still able to undergo fission and fusion (Fig. 1d), but improperly segregated into daughter cells after mitosis (Fig. 1e). This phenotype of defective mitochondrial segregation is consistent with the previous finding that spherical/giant mitochondria in mutant cells compromise mitochondrial movements, inheritance, and segregation[7,8,21]. Based on above observations, we concluded that Emr1 plays a critical role in regulating mitochondrial morphology.

**Emr1 is a transmembrane protein localized to the mitochondrial outer membrane.** Next, we examined the intracellular localization of Emr1. Emr1 was tagged either at the N-terminus or at the C-terminus with GFP (green fluorescent protein) and was expressed at the *leu1* locus from the *nmt41* promoter in *emr1Δ* cells (supplementary Fig. 1). Thiamine at different concentration was used to titrate the expression levels of Emr1. Intriguingly, Emr1-GFP (i.e., tagged at the C-terminus) and GFP-Emr1 (i.e., tagged at the N-terminus) showed completely different localization with the former one localizing to the ER and the latter one localizing to mitochondria (supplementary Fig. 1a, e). At low expression levels (i.e., expressed under the control of 0.015 μM or 0.03 μM thiamine), GFP-Emr1, but not Emr1-GFP, rescued the mitochondrial phenotype caused by the absence of Emr1. We noticed that Emr1 expressed from its own promoter displayed low expression levels (Fig. 2a, c). Therefore, we considered that GFP-Emr1, but not Emr1-GFP, is functional, though it is puzzling that Emr1-GFP expressed at high levels was also able to rescue the mitochondrial phenotype caused by the absence of Emr1 (supplementary Fig. 1a–c). In this present study, we used GFP-Emr1 to display the localization of Emr1.

As shown in Fig. 2a, c, the expression level of GFP-Emr1 that was expressed from its own promoter was very low, and accordingly only a few GFP-Emr1 foci were visible on mitochondria. Nonetheless, the mitochondrial phenotype in *emr1Δ* cells was rescued by the ectopic expression of GFP-Emr1. To better visualize the localization of GFP-Emr1, we then increased the expression level of GFP-Emr1 by using the *ase1* promoter (Fig. 2a, c). We found that GFP-Emr1 now not only displayed foci on mitochondria but also showed signals along mitochondria (Fig. 2a and Supplementary Fig. 2). When the expression level of GFP-Emr1 was further elevated by using the *nmt41* promoter, we observed quite bright signals of GFP-Emr1 on the mitochondrial outer membrane (Fig. 2b, c). Hence, these results suggest that Emr1 is a mitochondrial protein, and it likely localizes to the mitochondrial outer membrane.

Analysis of Emr1 amino acid sequences with the online bioinformatics tool TMHMM 2.0 (http://www.cbs.dtu.dk/services/TMHMM/)[28] predicted that Emr1 is a transmembrane protein

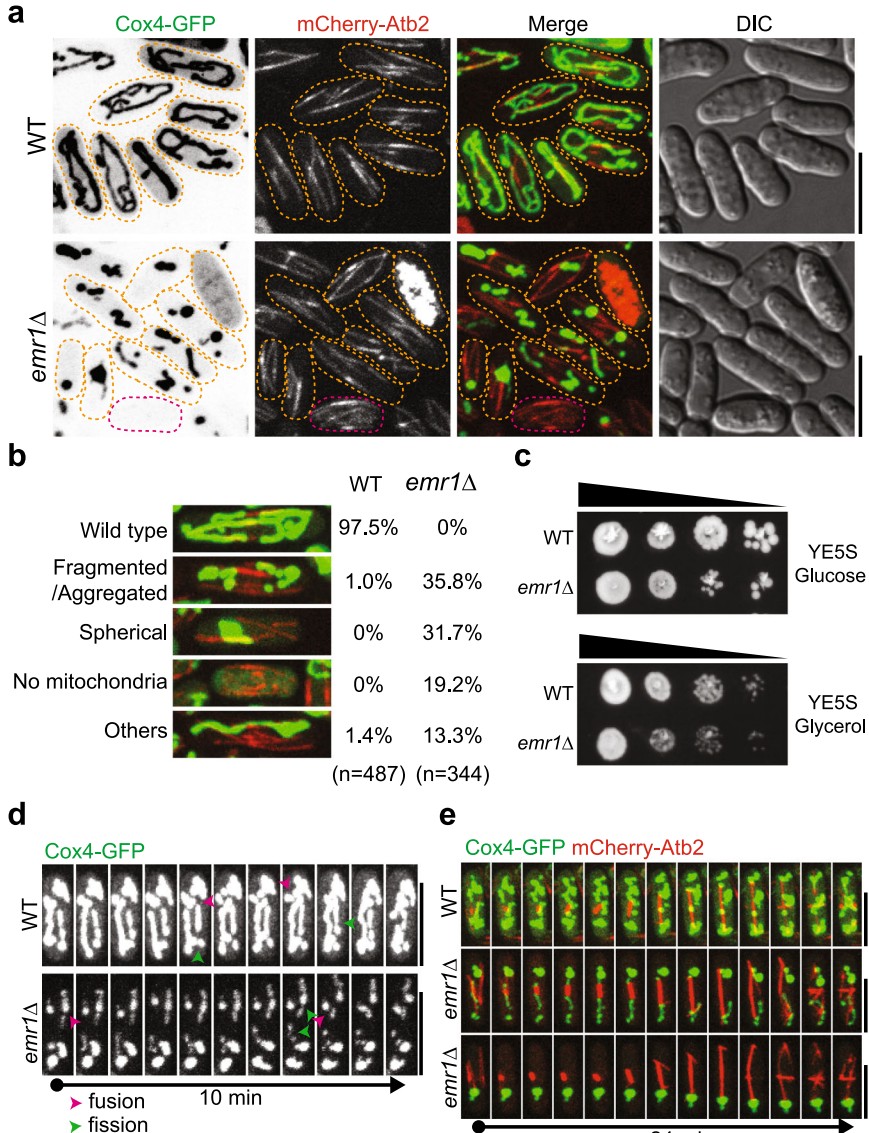

**Fig. 1 The absence of Emr1 leads to abnormal mitochondrial morphology. a** Maximum projection images of wild type (WT) and *emr1*Δ (*emr1*-deletion) cells expressing Cox4-GFP (a mitochondrial matrix protein) and mCherry-Atb2 (α-tubulin). Dashed lines mark the edge of cells and the red dashed line marks a cell lacking mitochondria. Note that mitochondria were tubular in WT cells but became spherical and/or aggregated in *emr1*Δ cells. Scale bar, 10 μm. **b** Quantification of mitochondrial phenotypes. On the left are representative images showing the indicated mitochondrial phenotypes. *n* indicates cell number observed for quantification. **c** Growth assays for WT and *emr1*Δ cells. The indicated cells were spotted on YE5S plates containing glucose (top) and YE5S plates containing glycerol (lower) after tenfold serial dilution to assess their growth on fermentable (glucose) and non-fermentable (glycerol) media, respectively. **d** Maximum projection time-lapse images of WT and *emr1*Δ cells expressing Cox4-GFP. Pink and green arrowheads mark mitochondrial fusion and fission, respectively. Scale bar, 10 μm. **e** Maximum projection time-lapse images of WT and *emr1*Δ cells expressing Cox4-GFP and mCherry-Atb2. Note that the cells were undergoing mitosis and one of the *emr1*Δ cells (bottom) segregated all mitochondria to only one end of the cell. Scale bar, 10 μm.

carrying a transmembrane domain consisting of 23 amino acid residues (a.a., 21–43) in the middle and that N-terminal (a.a., 1–20) and C-terminal (a.a., 44–61) regions localize inside and outside of the membrane, respectively (Fig. 2d). To test the prediction, we created an *emr1*Δ *yta4*Δ strain expressing Yta4-13Myc (Msp1 in budding yeast, containing an N-terminal transmembrane domain and localizing to the mitochondrial outer membrane)[29] and GFP-Emr1 (from the *ase1* promoter), and the topological structure of Emr1 on mitochondria was analyzed by carbonate extraction and proteinase K digestion assays using isolated mitochondria (Fig. 2e, f). As shown in Fig. 2e, similar to Yta4-13Myc, GFP-Emr1, but not the mitochondrial matrix protein Mti2, was present in the insoluble membrane fractions after carbonate extraction treatment. This

result suggests that Emr1 is an integral membrane protein. Intriguingly, the size of GFP-Emr1 became smaller upon digestion with proteinase K in the isotonic buffer, which was likely due to the degradation of the cytoplasmic C-terminus of Emr1 (Fig. 2f). In addition, the digestion of GFP-Emr1 by proteinase K was more efficient in the hypotonic buffer with and without Triton X-100, which was likely due to the rupture of the mitochondrial outer membrane and the release of the protein from the membrane, respectively (Fig. 2f). Consistently, due to the presence of the Myc tag in the cytoplasmic side, Yta4-13Myc was digested in all conditions by proteinase K, and the mitochondrial matrix protein Mti2 was digested only in the presence of Triton X-100. Collectively, these results support that Emr1 localizes to the

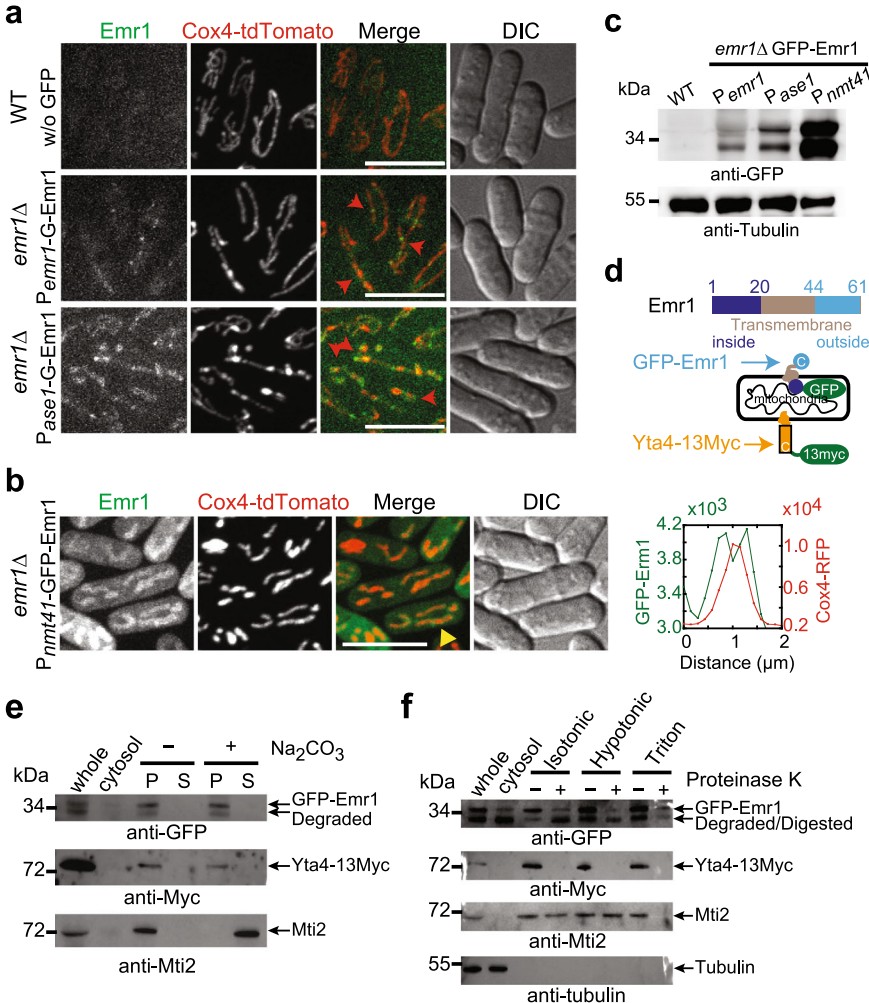

**Fig. 2 Emr1 is a mitochondrial outer membrane protein. a** Maximum projection images of WT cells expressing only Cox4-tdTomato or *emr1Δ* cells expressing Cox4-tdTomato and GFP-Emr1 either from *emr1*'s own promoter (P*emr1*) or an *ase1* promoter (P*ase1*). Emr1 localized along mitochondria and enriched at some places as foci (red arrows) (See supplementary Fig. 2 for all Z-stack images). Scale bar, 10 µm. **b** Maximum projection images of *emr1Δ* cells expressing Cox4-tdTomato and GFP-Emr1 from the *nmt41* promoter (P*nmt41*) in EMM5S medium without thiamine. Line-scan intensity measurement was performed perpendicularly along the mitochondria marked by the yellow arrowhead and the measurement data is shown as a plot graph on the right of the image. Scale bar, 10 µm. **c** Western blotting assays. The expression levels of GFP-Emr1 as shown in **a** and **b** were analyzed by western blotting with antibodies against GFP and tubulin, respectively. **d** Domain structure of Emr1. A diagram illustrates the localization of GFP-Emr1 and Yta4-13Myc (Msp1 in budding yeast, containing a transmembrane domain at the N-terminus) on the mitochondrial outer membrane. *C* indicates the C-termini of proteins. **e** Carbonate extraction of isolated mitochondria from cells expressing GFP-Emr1 and Yta4-13Myc. *whole* and *cytosol* indicates whole-cell lysate and the cytosol fraction from mitochondria isolation, respectively; *P* and *S* indicate insoluble membrane and soluble fractions, respectively; − and + indicate the absence and presence of sodium carbonate, respectively. Similar to Yta4-13Myc, GFP-Emr1 is present in the insoluble membrane fractions. Mti2 is a mitochondrial matrix protein. Western blotting assays were performed with antibodies against GFP, Myc, and Mti2, respectively. **f** Proteinase K digestion assays. *whole* and *cytosol* indicate whole-cell lysate and the cytosol fraction from mitochondria isolation, respectively. Isolated mitochondria from cells expressing GFP-Emr1 and Yta4-13Myc were incubated in the isotonic buffer (indicated as *Isotonic*), hypotonic buffer (indicated as *Hypotonic*), or hypotonic buffer containing 1% Triton X-100 (indicated as *Triton*), followed by digestion with proteinase K. The absence and presence of proteinase K are indicated by − and +, respectively. Western blotting assays were performed with antibodies against GFP, Myc, Mti2, and tubulin, respectively.

mitochondrial outer membrane with its C-terminus exposed in the cytoplasm as predicted in Fig. 2d. We also noticed that the cytoplasmic region of Emr1 may be prone to degradation because a smaller size of GFP-Emr1 from the whole-cell lysates and the cytosol fraction, obtained from mitochondria isolation, was detected (Fig. 2f).

Based on the topological structure of Emr1, we hypothesized that removal of the N-terminal or C-terminal region of Emr1 would not affect the localization of Emr1 but the absence of the C-terminus would be detrimental to the function of Emr1. To test this hypothesis, we first examined the localization of GFP-Emr1-ΔC (lacking the C-terminus) and GFP-Emr1-ΔN (lacking the N-

terminus). As shown in Fig. 3a, both truncated mutants localized to mitochondria, confirming the hypothesis that the transmembrane domain is required for the localization of Emr1 to mitochondria. We then assessed the effects of the Emr1 truncated mutants on mitochondrial morphology. As shown in Fig. 3b, c, similar to Emr1-FL (full-length), Emr1-ΔN, but not Emr1-ΔC, rescued the mitochondrial phenotype caused by the absence of Emr1. Consistently, our growth assays showed that *emr1-ΔC* and *emr1Δ* compromised cell growth to a similar extent (Fig. 3d). Remarkably, Mco6, the Emr1 ortholog in budding yeast, fully rescued the defective mitochondrial morphology in *emr1Δ* cells (Fig. 3b, c). Hence, Emr1 is a *bona fide* mitochondrial outer

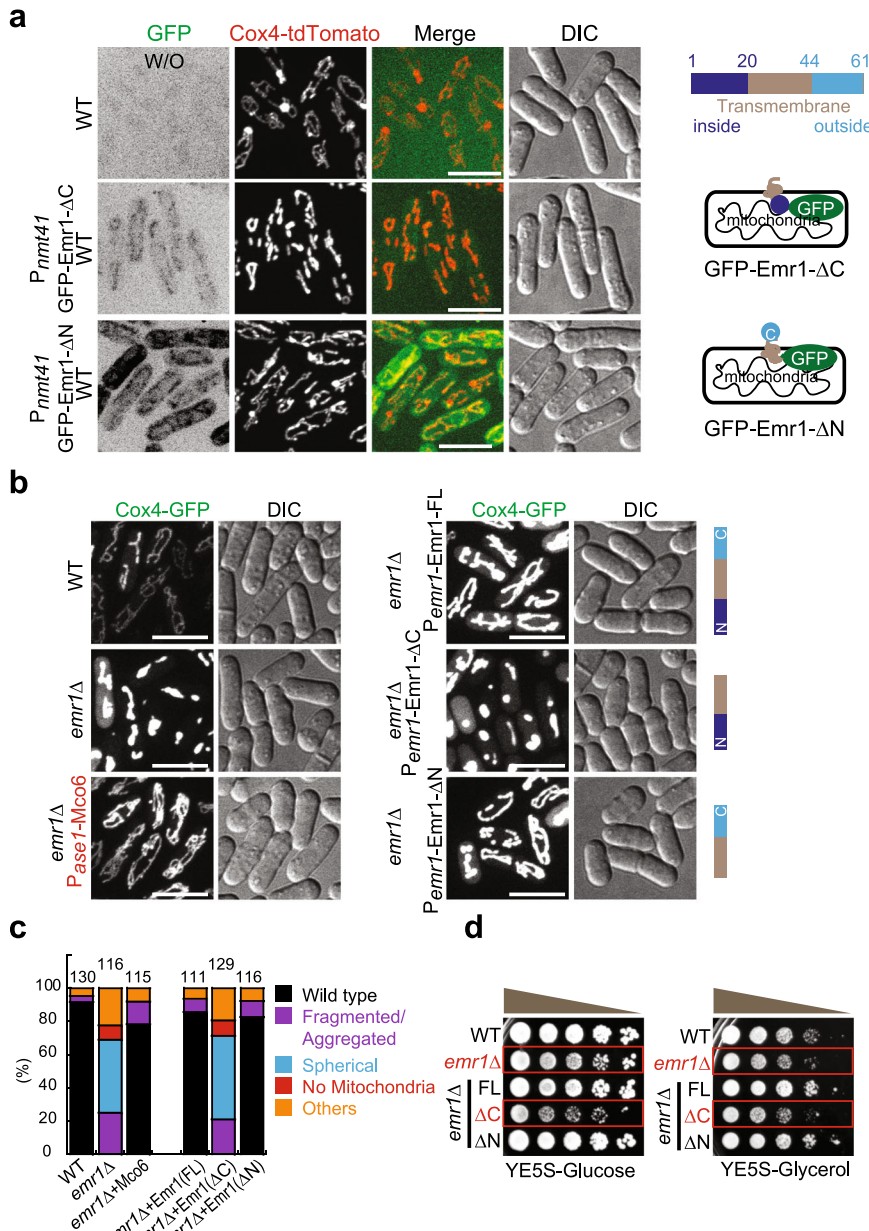

**Fig. 3 The C-terminus of Emr1 is required for maintaining proper mitochondrial morphology. a** Maximum projection images of WT cells expressing Cox4-tdTomato and either GFP-Emr1-ΔC or GFP-Emr1-ΔN (from the *nmt41* promoter). Note that cells expressing only Cox4-tdTomato did not display signals in the GFP channel (top panel). Domain structures of GFP-Emr1-ΔC (lacking the C-terminus (a.a. 44–61)) and GFP-Emr1-ΔN (lacking the N-terminus (a.a. 1–20)) are shown on the right. Scale bar, 10 μm. **b** Maximum projection images of the indicated cells expressing Cox4-GFP. Pase1-Mco6 (highlighted in red), the budding yeast homolog of Emr1 expressed from the *ase1* promoter; P*emr1*-Emr1-FL, full-length Emr1 expressed from its own promoter; P*emr1*-Emr1-ΔC, Emr1 lacking the C-terminus (a.a. 44–61) expressed from its own promoter; P*emr1*-Emr1-ΔN, Emr1 lacking the N-terminus (a.a. 1–20) expressed from its own promoter. Note that Mco6, Emr1-FL, and Emr1-ΔN, but not Emr1-ΔC, were able to rescue the mitochondrial phenotypes caused by the absence of Emr1. Scale bar, 10 μm. **c** Quantification of the indicated mitochondrial phenotypes for the cells in **b**. Cell number observed for quantification is shown on the top of the graph. **d** Growth assays for the indicated cells. The indicated cells were spotted on YE5S plates containing glucose and YE5S plates containing glycerol after tenfold serial dilution. These results are in agreement with the imaging data shown in **b**, i.e., *Emr1Δ* and Emr1-ΔC cells displayed impaired cell growth (highlighted by red rectangles).

member protein and its C-terminus plays a critical role in dictating Emr1 function.

**Emr1 regulates the number of ERMES foci.** It has been reported that malfunctions of the ERMES complex or mitochondrial outer membrane proteins lead to abnormal mitochondrial morphology, such as spherical and globular shapes[5,21], which is similar to those observed in *emr1Δ* cells. This prompted us to ask whether Emr1 is involved in regulating ERMES functions. As shown in Fig. 4a, b, the absence of Emr1 significantly decreased the number of Mdm12 (a constitutive component of the ERMES complex) foci (1–2 vs. 5–7 Mdm12 foci in WT cells). In some extreme cases, *emr1Δ* cells lost Mdm12 foci completely (Fig. 4a, b). We noticed that the remained Mdm12 foci in *emr1Δ* cells appeared to be

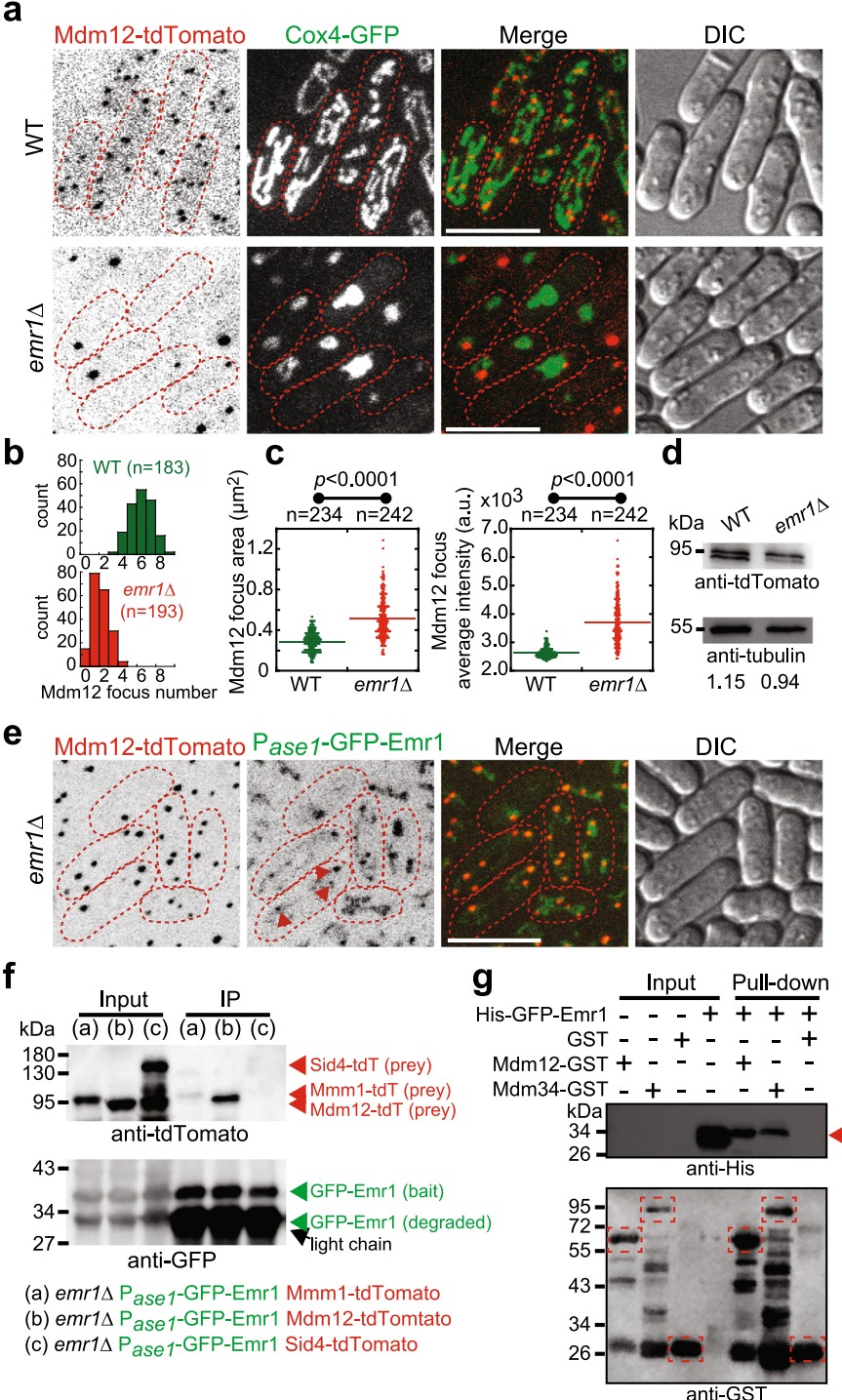

larger and brighter (Fig. 4a, c), though the expression levels of Mdm12 were comparable in WT and *emr1Δ* Cells (Fig. 4d). Similar results were obtained when Mmm1, another constitutive component of the ERMES complex, was analyzed (Fig. 5a–d). Altogether, these results suggest that Emr1 plays a crucial role in regulating the number of ERMES foci within the cell.

Furthermore, our confocal microscopic data showed that Emr1 colocalizes with Mdm12 and Mmm1 (Figs. 4e and 5e) and co-immunoprecipitation assays showed that Emr1 interacts with Mdm12 (Fig. 4f). It appeared that GFP-Emr1 did not completely co-localized with Mdm12 and some GFP-Emr1 signals were present on mitochondria (Fig. 4e). This is likely due to the higher

expression levels of GFP-Emr1 from the *ase1* promoter than from its own promoter (Fig. 2a, c). To further test the interaction between Emr1 and the ERMES complex, we generated recombinant proteins of the ERMES components, and successfully purified Mdm12-GST and Mdm34-GST, but not GST-Mmm1 (and its variants lacking the transmembrane domain). Using the recombinant ERMES proteins and His-GFP-Emr1, GST pull-down assays were performed and the result showed that Mdm12-GST and Mdm34-GST, but not GST alone, physically interact with His-GFP-Emr1 (Fig. 4g). Collectively, these data support the conclusion that Emr1 interacts with the ERMES complex and regulates the number of ERMES foci.

**Fig. 4 Emr1 interacts with Mdm12 and is required for regulating the number of ERMES foci. a** Maximum projection images of WT and emr1Δ cells expressing Mdm12-tdTomato (a component of the ERMES complex) and Cox4-GFP. Dashed lines mark the edge of cells. Scale bar, 10 μm. **b** Histogram of the quantification of Mdm12-tdTomato focus number for WT and emr1Δ cells indicated in **a**. n indicates the number of cells observed for the quantification. **c** Quantification of Mdm12-tdTomato area and average intensity, respectively, in WT and emr1Δ cells. Two-sided Student's t-test was used to calculate p-values (area, p < 0.0001; intensity, p < 0.0001). n indicates the number of foci observed for quantification. **d** Western blotting analysis of the expression levels of Mdm12-tdTomato in WT and emr1Δ cells. Antibodies against tdTomato and tubulin were used. Number below is the intensity ratio of tdTomato over tubulin. **e** Maximum projection images of emr1Δ cells expressing Mdm12-tdTomato and GFP-Emr1 (from the ase1 promoter). Dashed lines mark the edge of cells. The foci of GFP-Emr1 co-localized with Mdm12-tdTomato (indicated by red arrowheads) (also see supplementary Fig. 3 for all Z-stack images). Note that the number of Mdm12-tdTomato foci significantly increased in the cells (vs. emr1Δ cells in **a**). Scale bar, 10 μm. **f** Co-immunoprecipitation assay (Co-IP) to test the interaction between GFP-Emr1 and the ERMES components Mmm1-tdTomato and Mdm12-tdTomato. Sid4-tdTomato was used as negative control. The antibody against GFP was used to co-precipitate GFP-Emr1 in respective samples. GFP-Emr1 bands are indicated with green arrowheads while tdTomato fusion proteins are marked with red arrowheads. Western blotting assays were performed with antibodies against GFP and tdTomato. **g** GST pull-down assays were performed to test the interaction between the indicated recombinant His-tagged and GST-fused proteins. Note that GST-Mdm12 and GST-Mdm34, but not GST, physically interact with His-GFP-Emr1, which is marked by the red arrowhead. Dashed rectangles mark GST proteins.

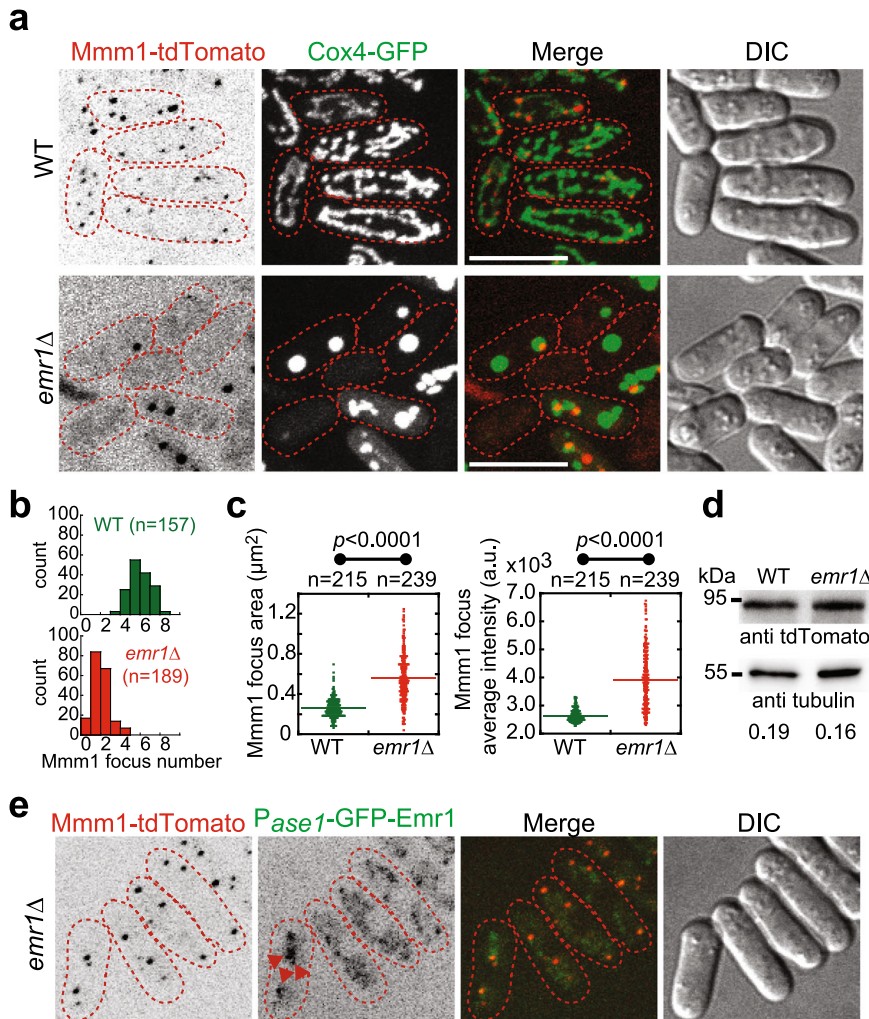

**Fig. 5 Emr1 colocalizes with Mmm1 and regulates the number, size, and average intensity of Mmm1 foci. a** Maximum projection images of WT and emr1Δ cells expressing Mmm1-tdTomato (a component of the ERMES complex) and Cox4-GFP. Dashed lines mark the edge of cells. Scale bar, 10 μm. **b** Histogram of the quantification of Mmm1-tdTomato focus number for WT and emr1Δ cells indicated in **a**. n indicates the number of cells observed for the quantification. **c** Quantification of Mmm1-tdTomato area and average intensity, respectively, in WT and emr1Δ cells. Two-sided Student's t-test was used to calculate p-values (area, p < 0.0001; intensity, p < 0.0001). n indicates the number of foci observed for quantification. **d** Western blotting analysis of the expression levels of Mmm1-tdTomato in WT and emr1Δ cells. Antibodies against tdTomato and tubulin were used. Number below is the intensity ratio of tdTomato over tubulin. **e** Maximum projection images of emr1Δ cells expressing Mmm1-tdTomato and GFP-Emr1 (from the ase1 promoter). GFP-Emr1 foci co-localized with Mmm1-tdTomato (indicated by red arrowheads). Dashed lines mark the edge of cells. Note that the number of Mmm1-tdTomato foci significantly increased in the cells (vs. the one in emr1Δ cells). Scale bar, 10 μm.

**The C-terminus of Emr1 is required for regulating the number of ERMES foci.** Based on the above findings (Figs. 2–5), we assumed that the cytoplasmic region of Emr1 (i.e., the C-terminus) might be responsible for regulating the number of the ERMES complex. To test this possibility, we expressed Emr1-FL (full-length), Emr1-ΔN (lacking the N-terminus), and Emr1-ΔC (lacking the C-terminus) from the *emr1* promoter in *emr1Δ* cells carrying Mdm12-tdTomato and Cox4-GFP (see diagrams for the

truncated mutants in Fig. 6a). As shown in Fig. 6b, c, Emr1-FL and Emr1-ΔN, but not Emr1-ΔC, restored the normal number of Mdm12 foci, confirming that the C-terminus of Emr1 is required for regulating the number of ERMES foci. Consistently, Emr1-FL and Emr1-ΔN, but not Emr1-ΔC, restored the normal mitochondrial morphology in *emr1Δ* cells (Fig. 3b and Supplementary Fig. 5a). In addition, expression of Mco6, the Emr1 homolog in budding yeast, was able to partially rescue the decreased number

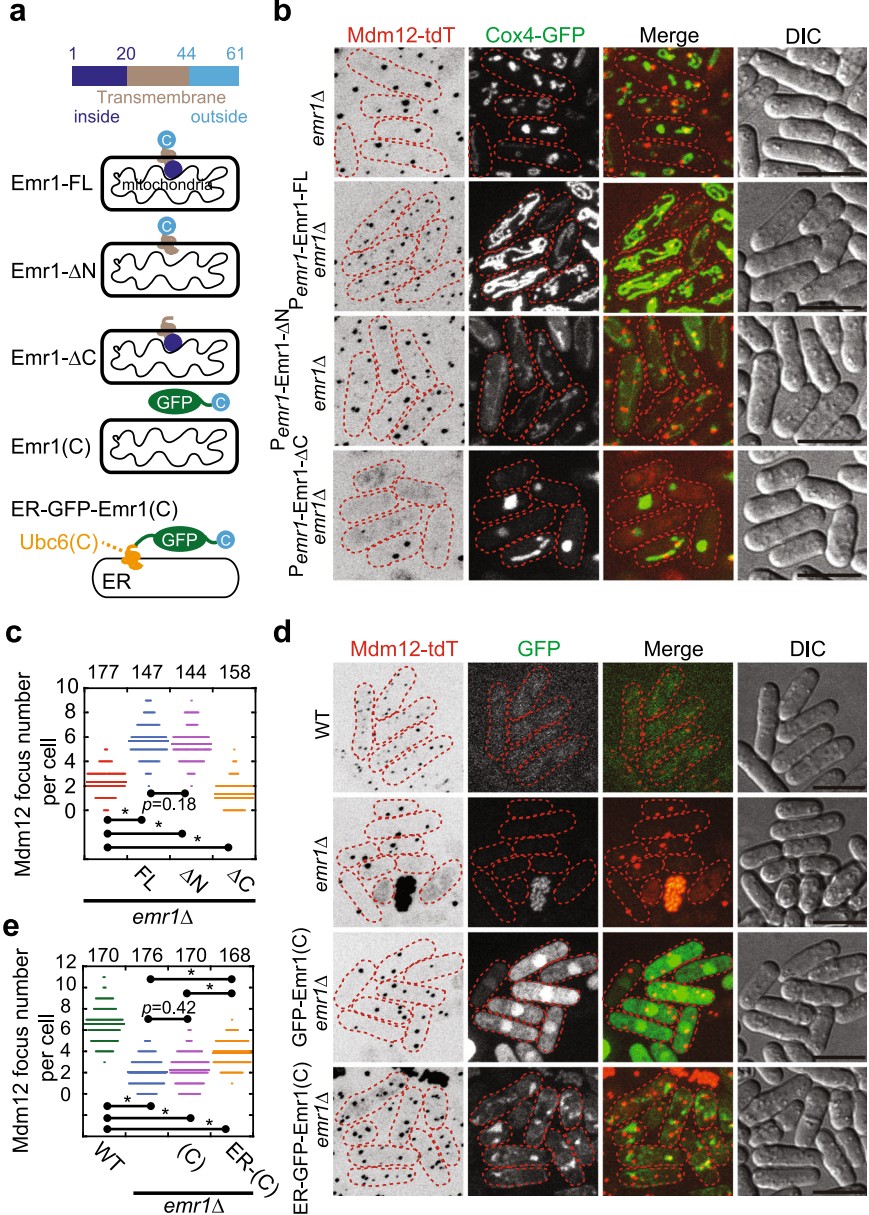

**Fig. 6 The C-terminus of Emr1 is required for regulating the number of ERMES foci. a** Diagrams illustrating the different modes of localization of the indicated Emr1 truncated mutants and the chimera ER-GFP-Emr1(C). *ER* is the ER-localizing fragment from Ubc6 (including its C-terminus (a.a. 226–227) and the adjacent transmembrane domain (a.a. 207–225)). **b** Maximum projection images of *emr1Δ* cells expressing Mdm12-tdTomato, Cox4-GFP, and the indicated Emr1 proteins (from the *emr1* promoter). Both full-length and ΔN, but not ΔC, of Emr1 were able to restore the normal number of Mdm12-tdTomato foci. Dashed lines mark the edge of cells. Scale bar, 10 μm. **c** Quantification of Mdm12-tdTomato foci in the indicated cells in **b**. Statistical analysis was performed by one-way ANOVA ($F(3, 622) = 530.26$, $p < 0.001$), followed by Tukey honest significance difference test (the *p*-values are indicated on the graph; $*p < 0.001$). *n* indicates cell number observed for quantification. **d** Maximum projection images of the indicated cells expressing Mdm12-tdTomato and the indicated GFP-fused proteins. Emr1(C), the C-terminus of Emr1; ER-Emr1(C), an ER-targeting peptide fused to the N-terminus of the cytoplasm region of Emr1. Note that ER-Emr1(C), but not Emr1(C), was able to partially rescue the number of Mdm12 foci. Dashed lines mark the edge of cells. Scale bar, 10 μm. **e** Quantification of Mdm12-tdTomato foci in the indicated cells in **d**. Statistical analysis was performed by one-way ANOVA ($F(3, 680) = 373.86$, $p < 0.001$), followed by Tukey honest significance difference test (the *p*-values are indicated on the graph; $*p < 0.001$). *n* indicates cell number observed for quantification.

of ERMES foci in *emr1Δ* cells (supplementary Fig. 4c–e). The partial rescue may be due to the low amino sequence similarity (~30% identity by alignment) and the short C-terminus of Mco6, as compared with Emr1 (supplementary Fig. 4e). We then asked if the C-terminus alone is enough for restoring the normal number of Mdm12 foci in *emr1Δ* cells. The Emr1 C-terminus alone (referred to as Emr1(C)) localized homogeneously within the cell and did not increase the number of Mdm12 foci in *emr1Δ* cells (Fig. 6d, e). These results suggest that the Emr1 C-terminus alone is not sufficient for determining the number of ERMES foci.

We further asked if localizing Emr1(C) to the ER, instead of mitochondria, is able to restore the number of ERMES foci in *emr1Δ* cells. To test this idea, we generated a chimera protein by fusing an ER-localizing fragment from Ubc6 (including its C-terminus and the adjacent transmembrane domain) with GFP-Emr1(C), Emr1(C), and 13Myc-Emr1(C), which were named as ER-GFP-Emr1(C), ER-Emr1(C), and ER-13Myc-Emr1(C), respectively. ER-GFP-Emr1(C) localized within the cytoplasm and to the ER (Fig. 6d), and only slightly restored the number of Mdm12 foci in *emr1Δ* cells (Fig. 6d, e). Similarly, ER-Erm1(C) (without a linking region between the ER module and Emr1(C)) and ER-13Myc-Em1(C) (a shorter linking region is present between the ER module and Emr1(C)) also only partially restored the number of Mdm12 foci in *emr1Δ* cells (Supplementary Fig. 4a). Therefore, tethering Emr1(C) to the ER is not sufficient for maintaining proper number of ERMES foci within the cell. Surprisingly, tethering Emr1(C) to mitochondria by fusing the transmembrane domain of Yta4 to Emr1(C) did not fully restored the number of Mdm12 foci in *emr1Δ* cells, though appeared to function better than the chimeras targeting to the ER (supplementary Fig. 4b). Hence, we concluded that the cytoplasmic region of Emr1 plays an important role in regulating the number of ERMES foci but both the cytoplasmic region and the transmembrane domain are required for Emr1 to be fully functional.

**Artificial tethering of mitochondria to the ER restores normal mitochondrial morphology but not the number of ERMES foci in *emr1Δ* cells**. Malfunction of the ERMES complex impairs the contact between the ER and mitochondria, leading to abnormal mitochondrial morphology[3,5,7,8,30]. To assess if the abnormal mitochondrial morphology observed in *emr1Δ* cells is directly due to the malfunction of the ERMES complex, we employed the chimera tether developed previously for tethering the ER to mitochondria (Fig. 7a)[5]. As shown in Fig. 7b, c, expression of the chimera tether in *emr1Δ* cells did not restore the number of ERMES foci. By contrast, expression of the chimera tether was able to restore the tubular mitochondria network in *emr1Δ* cells (Fig. 7d, e and supplementary Fig. 5b). In addition, the occupancy of mitochondria within the cell increased significantly in the tether-expressing *emr1Δ* cells, indicative of restoration of mitochondrial mass (Fig. 7e). Collectively, these results suggest that the abnormal mitochondrial morphology observed in *emr1Δ* cells is directly due to compromised contacts between the ER and mitochondria caused by the absence of Emr1. Additionally, the results suggest that the decreased number of ERMES foci observed in *emr1Δ* cells is not a consequence of the abnormal mitochondrial morphology, highlighting the critical role of Emr1 in regulating the number of ERMES foci.

Given the role of the ERMES complex in mediating phospholipid transfer between the ER and mitochondria[3,12,13], we further asked whether the absence of Emr1 affects lipid transfer between the ER and mitochondria. To address this question, we employed the approach of thin-layer chromatography (TLC), developed previously[31], to analyze the amount of phosphatidylethanolamine

(PE), phosphatidylserine (PS), and cardiolipin (CL) in whole-cell lysates and isolated mitochondria of WT and *emr1Δ* cells. As shown in Fig. 8a–d, PE was significantly reduced in the absence of Emr1. Moreover, although the reduction of PS in mitochondria of *emr1Δ* cells was not statistically significant, the PS ratios of *emr1Δ* over WT, measured from four independent experiments, were 0.85, 0.71, 1.08, and 0.89, respectively (Fig. 8d). This result indicates a likely reduction of PS in mitochondria in the absence of Emr1. Since PE is mainly synthesized within mitochondria by converting PS that is transferred from the ER, the reduced amount of PE in mitochondria likely suggests that Emr1 is involved in regulating phospholipid transfer between the ER and mitochondria, presumably through the ERMES complex. Although PE is mainly synthesized within mitochondria by the PS decarboxylase Psd1, PE could be generated by the Kennedy pathway (Ept1 in the pathway was removed to block the pathway in this present study) and by Psd2 and Psd3 in other synthesis routes[32,33]. Intriguingly, *emr1Δ* and *psd1Δ* did not have an additive effect on cell growth but *emr1Δ* had an additive effect on cell growth if combined with *psd2Δ*, *psd3Δ*, and *ept1Δ* (Fig. 8e). Altogether, these results suggest that Emr1 is involved in regulating PE synthesis, likely by regulating the phospholipid transfer between the ER and mitochondria through the ERMES complex.

## Discussion

The ERMES complex functions as a tether at ER-mitochondria contact sites to regulate many important intracellular activities, including mitochondrial morphology and dynamics, lipid transport, mtDNA inheritance, and mitophagy[4]. The protein(s) responsible for regulating the number of ERMES foci remains to be determined. Here, we demonstrate that Emr1 is a regulatory protein of the ERMES complex and it plays crucial roles in determining the number of ERMES foci and regulating mitochondrial morphology (Figs. 1, 4, and 5).

We provide strong evidence to support that Emr1 serves as a crucial regulatory subunit of the ERMES complex. First, the absence of Emr1 significantly decreases the number of Mdm12 and Mmm1 foci (Figs. 4a, b and 5a, b). Second, Emr1 physically interacts with Mdm12 and Mdm34 (Fig. 4f, g). Third, Emr1 colocalizes with Mdm12 and Mmm1 (Figs. 4e and 5e). The recently identified peripheral subunits of the ERMES complex in budding yeast, i.e., Gem1 and Lam6, also colocalize and co-precipitate with the ERMES complex, but they do not appear to play a major role in regulating ERMES assembly[19,21,22]. Similarly, the recently identified binding protein of Mdm10, i.e., Tom7, functions as a nonessential regulatory subunit of the ERMES complex[24]. By contrast, the maintenance of ERMES focus number and the proper synthesis of PE largely depend on Emr1 (Figs. 4, 5, and 8a–d). Therefore, Emr1 is a crucial regulatory protein required for proper functions of the ERMES complex.

Emr1 likely has a specific role in regulating the number of ERMES foci because simply tethering the ER to mitochondria in *emr1Δ* cells with the artificial tether, developed previously[5], is able to restore tubular mitochondrial morphology (Fig. 7d, e) but does not restore the normal number of ERMES foci (Fig. 7b, c). Similarly, the artificial tether is also able to rescue defective mitochondrial morphologies in Mdm12 and Mdm34 mutant cells, but not in Mdm10 and Mmm1 mutant cells[5]. Mdm10 not only anchors the ERMES complex to mitochondria but also is involved in the functions of the SAM and TOM complexes on the mitochondrial outer membrane[24,34]. Mmm1 anchors the ERMES complex to the ER and regulates the assembly of β-barrel proteins on the mitochondrial outer membrane[35]. Therefore, it is likely that Mdm12 and Mdm34, but not Mmm1 and Mdm10, function specifically in the ERMES complex. Likewise, Emr1 functions

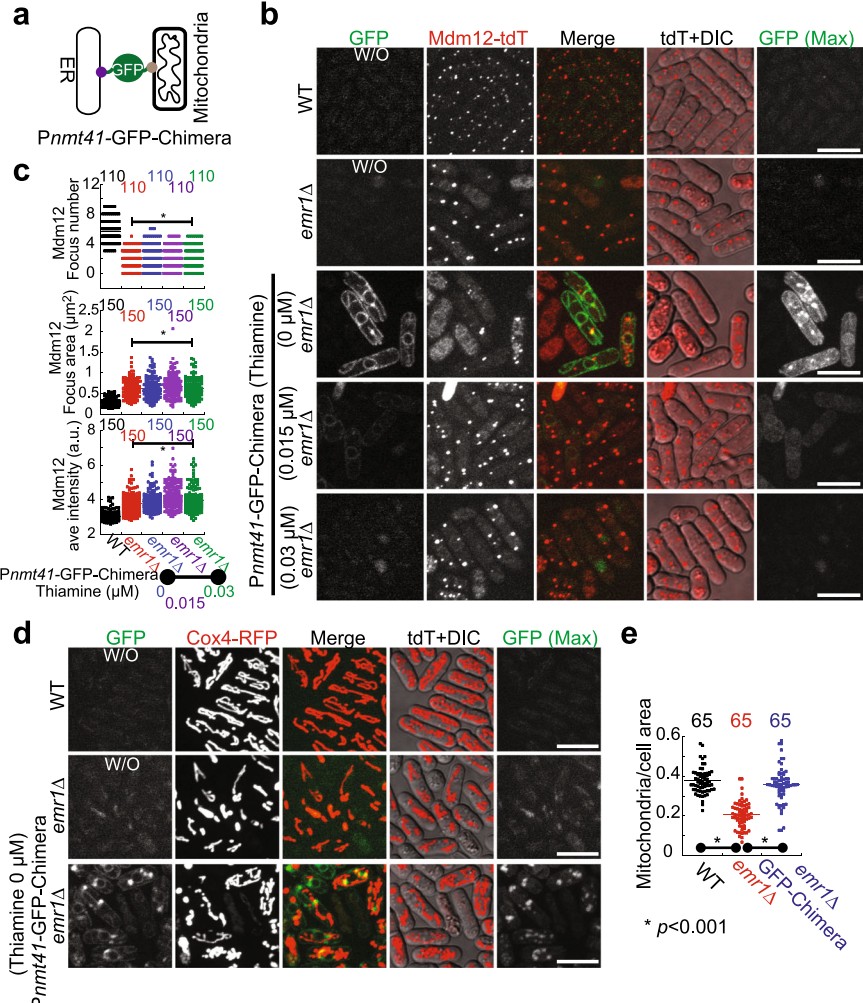

**Fig. 7 Tethering mitochondria to the ER partially restores normal mitochondrial morphology but not the number of ERMES foci in *emr1Δ* cells.**
**a** Diagram illustrating the design of the chimera tether, developed previously[5]. Two domains capable of inserting into the ER membrane and the mitochondrial outer membrane, respectively, are linked by GFP. **b** Maximum projection images of WT and *emr1Δ* cells expressing Mdm12-tdTomato. To assess the effect of the chimera tether on ERMES assembly, the chimera tether was expressed from the *nmt41* promoter in media containing 0, 0.015, and 0.03 μM thiamine (suppression of the tether expression), respectively, in *emr1Δ* cells. Note that images in the GFP panel are the middle plane of the respective Z-stack images. The strains that do not express GFP-tagged proteins are marked by W/O. Scale bar, 10 μm. **c** Quantification of Mdm12-tdTomato focus number, area and average intensity, respectively, in the indicated cells in **b**. Note that expression of the chimera tether does not rescue defective formation of the ERMES complex in *emr1Δ* cells. Statistical analysis was performed by one-way ANOVA (focus number: $F_{(4, 545)}=138.60$, $p < 0.0001$; focus area: $F_{(4, 745)}=73.83$, $p < 0.0001$; focus average intensity: $F_{(4, 745)} = 88.65$, $p < 0.0001$), followed by Tukey honest significance difference test (vs. WT; *, $p < 0.0001$). *n* indicates cell number observed for quantification. **d** Maximum projection images of the indicated cells expressing Cox4-RFP. Cells were cultured in EMM5S media without thiamine. At the bottom panel are the cells expressing the chimera tether. Expression of the tether largely restored normal mitochondrial morphology in *emr1Δ* cells. Note that images in the GFP panel are the middle plane of the respective Z-stack images. The strains that do not express GFP-tagged proteins are marked by W/O. Scale bar, 10 μm. **e** Quantification of the ratio of mitochondrial area and cell area. Mitochondrial area increases significantly in *emr1Δ* cells after ER-mitochondria contacts are rescued with the chimera tether. Statistical analysis was performed by one-way ANOVA ($F_{(2, 192)} = 100.10$, $p < 0.0001$), followed by Tukey honest significance difference test (*emr1Δ* vs. WT and *emr1Δ* vs. the chimera tether; *$p < 0.0001$). *n* indicates cell number observed for quantification.

specifically in regulating the ERMES complex. Malfunction of ERMES core components leads to ERMES disassembly, causing spherical/grape-like mitochondria[5,7–9,11]. It is, therefore, conceivable that the impaired mitochondrial morphology in *emr1Δ* cells is a direct effect of ERMES malfunction caused by the absence of Emr1.

Mechanisms underlying the assembly of the four core components of the ERMES complex remain unclear due to the lack of structural data for the entire ERMES complex. Nonetheless, the interaction between the two SMP domain-containing proteins Mdm12 and Mmm1 has been characterized structurally[10,12,13,36], showing the importance of the SMP domains in mediating the

intermolecular interaction. Additionally, the interaction of Mdm34 with the Mmm1-Mdm12 subcomplex appears to be weak[36]. How the β-barrel protein Mdm10 anchors the ERMES complex on the mitochondrial outer membrane is unknown.

How does Emr1 regulate the number of ERMES foci and/or the assembly of the ERMES complex? Our present data point to several possibilities. First, Emr1 may synergize with Mdm10 to anchor the ERMES complex since both proteins are inserted into the outer mitochondrial membrane (Fig. 2a–f)[5] and Emr1 physically interacts with Mdm34 and Mdm12 (Fig. 4g). The localization of Emr1 to mitochondria is determined by its transmembrane domain while its cytoplasmic domain, i.e., the

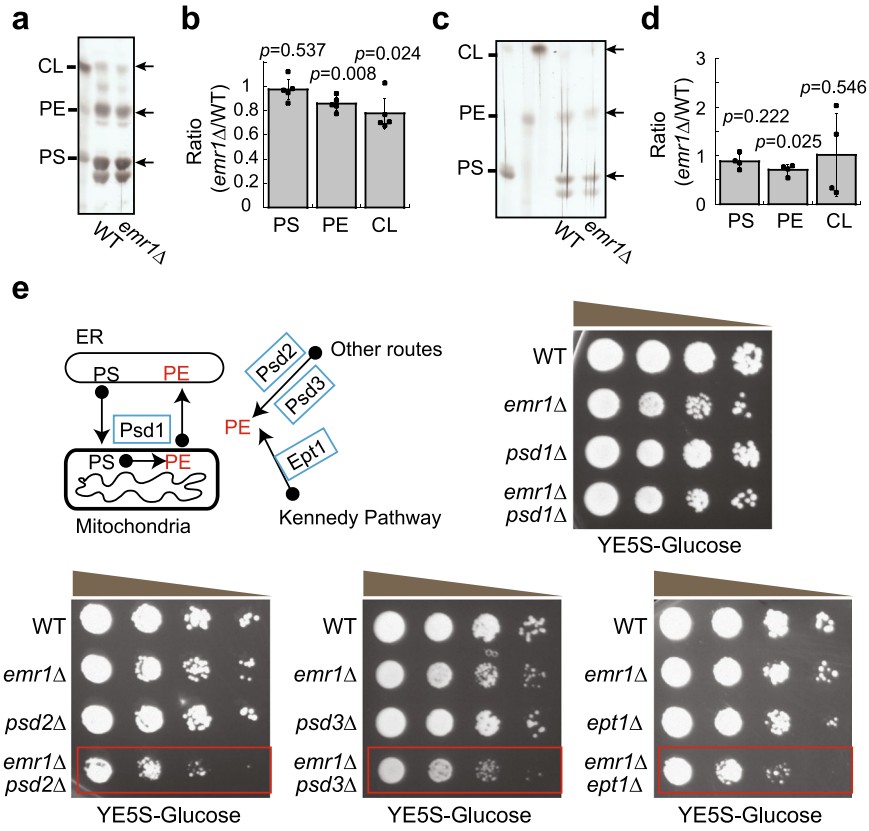

**Fig. 8 The absence of Emr1 affects the synthesis of phosphatidylethanolamine. a** Thin-layer chromatography (TLC) assays. TLC analysis of phospholipids extracted from WT and *emr1Δ* cells (whole-cell lysates). Phosphatidylserine (PS), phosphatidylethanolamine (PE), and cardiolipin (CL) were used as lipid standards (indicated by arrows). **b** Quantification of CL, PE, and PS shown in **a**. The band intensity of *emr1Δ* cells was normalized to the one of WT cells (displayed as Ratio). Data from five independent experiments were used for the quantification (indicated by dots). Error bars represent confidence intervals (95%), and the center of the error bar is the mean. Two-sided single-group Student's *t*-test was used to calculate the *p*-values of log ratio for each type of phospholipids (PS, *p* = 0.537; PE, *p* = 0.008; CL, *p* = 0.024). **c** TLC analysis of phospholipids extracted from isolated mitochondria of WT and *emr1Δ* cells. Arrows indicate lipid standards. **d** Quantification of CL, PE, and PS shown in **c**. The band intensity of *emr1Δ* cells was normalized to the one of WT cells (displayed as Ratio). Data from four independent experiments were used for the quantification (indicated by dots). Error bars represent confidence intervals (95%), and the center of the error bar is the mean. Two-sided Single-group Student's *t*-test was used to calculate the *p*-values of log ratio for each type of phospholipids (PS, *p* = 0.222; PE, *p* = 0.025; CL, *p* = 0.546). **e** Growth assays for the indicated cells. The indicated cells were spotted on YE5S plates containing glucose after tenfold serial dilution. Diagram showing the routes of PE synthesis. (1) The PS decarboxylase Psd1 resides in mitochondria to convert PS to PE after PS is transferred from the ER to mitochondria through the contact sites between the ER and mitochondria; (2) the PS decarboxylases Psd2 and Psd3 likely resides in other cellular compartments (except mitochondria) to synthesize PE; (3) Ept1 in the Kennedy pathway promotes the synthesis of PE. Note that *psd2Δ*, *psd3Δ*, and *ept1Δ*, but not *psd1Δ*, had an additive growth defect with *emr1Δ* (highlighted by red rectangles).

C-terminus, is required for regulating the number of ERMES foci (Figs. 3a and 6b, c). Artificial targeting of Emr1 C-terminus to the ER partially restores the number of Mdm12 foci suggests another possibility that membrane-localized Emr1 C-terminus facilitates ERMES assembly. It is also possible that Emr1 affects the association among the four core components of the ERMES complex given the physical interaction between Emr1 and Mdm12 and Mdm34 revealed in this present study (Fig. 4g). Further biochemical reconstitution with ERMES core components and Emr1 will help distinguish the possibilities.

The ERMES complex is not present in metazoans[37]. Nevertheless, its functional counterpart is present in mammals because the SMP domain-containing protein PDZD8 has been shown to be the functional homolog of Mmm1 and functions as an ER-mitochondria tether in mammalian cells[38,39]. Similarly, although Emr1 is conserved only in fungi (Fig. 3b–d and supplementary Fig. 4c–e), it is still possible that a functional counterpart of Emr1 is indeed present in mammals and functions to regulate ER-mitochondria tethers. This awaits further investigations.

## Methods

**Yeast genetics**. The yeast strains used in this study are given in Supplementary Table S1. The strains were created by random spore digestion or tetrad dissection[40]. The pFA6a series of plasmids were used for tagging and deletion of genes by the PCR-based method[41]. Briefly, the PCR amplified pFA6a cassettes were transformed into yeast cells by the Lithium acetate method. Cells were grown in EMM media (Edinburgh minimal media) containing the five supplements (0.225 g/l each): Adenine, Leucine, Uracil, Histidine, and Lysine (referred to as EMM5S), all purchased from Formedium (www.formedium.com) for imaging. To suppress the expression of genes controlled by the *nmt41* promoter, thiamine at the indicated concentration was added to EMM5S media. YE media having the five supplements (referred to as YE5S) were used to culture cells for biochemistry.

**Molecular cloning**. The restriction enzymes used for molecular cloning were purchased from NEB (www.neb.com). The ClonExpress II One Step Cloning Kit (www.vazymebiotech.com) was used to generate different truncation mutants of Emr1.

For creation of the plasmid GFP-Emr1(C) (pCF.3285), the transmembrane region (a.a. 21–43) and the N-terminus (a.a. 1–20) of *emr1* from the plasmid Pnmt41-GFP-Emr1 (pCF.3178) was deleted and replaced with a GSx5 (encoded by GGA AGT GGA AGT GGA AGT GGA AGT GGA AGT) linker by the in vitro recombination method (according to the user manual of ClonExpress II One Step

Cloning Kit), using oligoes oCF.3476 and oCF.3477 (see Supplementary Table S3 for details).

For creation of the chimera plasmid ER-GFP-Emr1(C) (pCF.3302) (see diagram in Fig. 6a), The C-terminus (a.a. 226–227) and the transmembrane region (a.a. 207–225) of *ubc6* was integrated, by the in vitro recombination method, at the upstream of GFP of the plasmid *Pnmt41*-GFP-Emr1(C) (pCF.3285), using oligoes oCF.3488 and oCF.3489.

For constructing the chimera tether (see diagram in Fig. 7a), the mitochondrial targeting peptides of Tom70 (a.a. 1–51, containing a transmembrane domain) was fused to the N-terminus of GFP while the ER-targeting peptides of Ubc6 (a.a. 207–227, containing a transmembrane domain) was fused to the C-terminus of GFP.

For creation of the chimera plasmid ER-Emr1(C) (pCF.3860) and ER-13Myc-Emr1 (C) (pCF.3861) (see diagram in Supplementary Fig. 4a), the GFP in ER-GFP-Emr1(C) (pCF.3302) was removed by using oCF.4251 and oCF.4252 and 13Myc was integrated to replace GFP in ER-GFP-Emr1(C) (pCF.3302) by the in vitro recombination method, respectively.

To create Yta4(TM)-GFP-Emr1(C) (pCF.3863) (see diagram in supplementary Fig. 4b), the N-terminal transmembrane domain of Yta4 (a.a. 1–38) was first cloned into pJK148-P*nmt41*-GFP plasmid to generate pJK148-P*nmt41*-Yta4(TM)-GFP, and then the C-terminus of Emr1 (i.e., Emr1(C)) was integrated downstream of the GFP in pJK148-Pnmt-Yta4(TM)-GFP by the in vitro recombination method, using oligoes oCF.4627 and oCF.4628.

All plasmids and oligoes used in this study are summarized in Supplementary Tables S2 and S3, respectively.

**Biochemistry.** Co-immunoprecipitation (Co-IP) assays were performed by incubating the cell lysates of strains expressing GFP- or tdTomato-tagged proteins with Dynabeads Protein G (www.thermofisher.com) bound with an anti-GFP antibody (raised by GenScript, www.genscript.com)[42]. Cell lysates were prepared by grinding in liquid nitrogen with a mortar grinder RM 200 (www.retsch.com) and dissolved in TBS lysis buffer containing 0.3% Triton X-100 and cocktail protease inhibitors. Dynabeads Protein G beads (www.thermofisher.com) bound with the anti-GFP antibody (raised by GenScript, www.genscript.com.cn) were incubated with the cell lysates for 1 h at 4 °C, followed by washing with 1x TBS buffer containing 0.1% Triton X-100 for five times and with 1x TBS buffer for one time. Co-IP products were then analyzed by western blotting with antibodies against GFP (dilution factor, 1:3000; www.rokland-inc.com) and tdTomato (dilution factor, 1:2000; www.origene.com).

For analysis of protein expression levels, protein extract was prepared by the NaOH lysis method[43]. Exponential cells were collected from 10 ml culture. After washed 1 time with 1 ml distilled deionized water (ddwater), the cells were resuspended in 0.5 ml ddwater, and then 0.5 ml NaOH (0.6 M) was added. The suspension was incubated at room temperature for 10 mins, and cells were collected by centrifugation at 6000 rpm for 30 s. The cell pellets were boiled in SDS sample buffer (60 mM Tris-HCl (pH 6.8), 4% SDS, 4% β-mercaptethanol, 5% glycerol, and 0.002% bromophenol blue) for 5 min and analyzed by western blotting with antibodies against GFP (dilution factor, 1:3000; www.rokland-inc.com) or tdTomato (dilution factor, 1:2000; www.origene.com) and tubulin (dilution factor, 1:3000; www.bioacademia.co.jp).

For GST pull-down assays, Mdm12-GST, Mdm34-GST, GST, and His-GFP-Emr1 were expressed in *E.coli* BL21 and were purified with Glutathione Sepharose 4B resins (www.gelifesciences.com) or nickel resins (www.qiagen.com). The purified GST proteins bound with Glutathione Sepharose 4B resins were incubated with equal amount of His-GFP-Emr1 eluted from nickel resins in 1X TBS buffer plus 0.1% Triton X-100 for 2 h at 4 °C. The GST-bound resins were then washed with 1X TBS plus 0.1% Triton X-100 and 10 mM imidazole for three times and with 1x TBS for one time. Finally, the GST-bound resins were dissolved in 1x SDS sample buffer and boiled for 10 mins at 100 °C. Sodium dodecyl sulfate–polyacrylamide gel electrophoresis (SDS–PAGE) and western blotting were performed to detect His-GFP-Emr1 with anti-His antibody (dilution factor, 1:2000; www.abclonal.com.cn) and the GST-fused proteins with anti-GST antibody (dilution factor, 1:2000; www.abclonal.com.cn).

Secondary antibodies used in this study are as follows: Goat Anti-Mouse-HRP Conjugate (dilution factor, 1:10,000; www.bio-rad.com), Goat Anti-Rabbit-HRP Conjugate (dilution factor, 1:10,000; www.bio-rad.com), and Rabbit Anti-Goat-HRP Conjugate (dilution factor, 1:10,000; www.abclonal.com.cn).

**Phospholipid extraction.** Fission yeast (*Schizosaccharomyces pombe*) cells were grown in YE5S media until mid-log phase and the same OD units (~15) of WT and *emr1Δ* cells were collected and washed 3 times with ddH$_2$O. Cell pellets were then dissolved in 400 μl of methanol and disrupted by vortex, using Retsch MM400 (www.retsch.com) with 100 μl of glass beads for 4–5 min. Vortex was performed again for 2–3 min after 800 μl chloroform was added. After the vortex, samples were centrifuged at 10,000 × g for 10 min to remove insoluble materials and supernatants were collected. To separate methanol from chloroform, 200 μl ddH$_2$O was added to the supernatant, and mixed thoroughly. After the separation, the methanol-containing layer on the top was removed and the lipid-containing chloroform layer at the bottom of the tube was collected carefully, followed by adding 400 μl 0.9% NaCl. A water-containing layer on the top of chloroform then

formed and was removed. Finally, the lipid-containing chloroform layer was collected and dried at 55 °C and were dissolved with 40 μl fresh chloroform.

For extraction of phospholipids from mitochondria, isolated mitochondria were used to replace the cell pellets described above. The final extracted phospholipids were dissolved in 20 μl fresh chloroform.

**Thin-layer chromatography (TLC).** Extracted lipids were analyzed by TLC on Silica Gel 60G Plates (www.sigmaaldrich.com) in a solvent mixture containing Chloroform, Acetone, Acetic Acid, Methanol, and Water (25:10:7.5:5:2.5, volume)[31]. The TLC plates were then dried, stained with 470 mM CuSO$_4$ in 8% o-phosphoric acid, and finally incubated at 160 °C for 15–20 min. Lipids standards (www.sigmaaldrich.com) were used to identify cardiolipin (CL), phosphatidylethanolamine (PE), and phosphatidylserine (PS). Metamorph 7.7 was used to measure band intensity.

**Emr1 topology.** *emr1Δyta4Δ* cells expressing Yta4-13myc (from its own promoter) and GFP-Emr1 (from the *ase1* promoter) were grown in YE5S media until OD$_{600}$ reached 0.8. Mitochondria were then isolated by using the Yeast Mitochondria Isolation Kit purchased from abcam (www.abcam.com) according to the user manual. For carbonate extraction, mitochondria were treated with sodium carbonate (100 mM Na$_2$CO$_3$, pH = 11.5) for 30 min on ice and subjected to ultra-centrifugation (100,000 × g for 30 min) to separate soluble and insoluble/membrane proteins. Soluble fractions were precipitated by 15% trichloroacetic acid (TCA). Proteins from soluble and insoluble fractions were analyzed by SDS–PAGE and western blotting with antibodies against GFP (dilution factor, 1:3000; www.rockland-inc.com), Myc (dilution factor, 1:2000; www.thermofisher.com), and Mti2 (a gift from Dr. Ying Huang at Nanjing Normal University; dilution factor, 1:2000)[44]. For proteinase K digestion assays, the experiments were conducted as described previously[30]. Briefly, isolated mitochondria were incubated in the isotonic SHE buffer (0.6 M Sorbitol, 20 mM HEPES, pH = 7.4) or hypotonic H buffer (20 mM HEPES, pH = 7.4) with and without 1% Triton X-100. Proteinase K (20 μg/μl) (www.tiangen.com) was then added and incubated on ice for 20 min. The digestion reaction was stopped with 2 mM PMSF and the mixtures were then incubated for 5 min on ice. After the stop reaction, the mixtures were boiled in 1x SDS sample buffer for 10 min at 100 °C and were analyzed by western blotting with the antibodies against GFP (dilution factor, 1:3000; www.rockland-inc.com), Myc (dilution factor, 1:2000; www.thermofisher.com), tubulin (dilution factor, 1:3000; www.bioacademia.co.jp), and Mti2. As control, whole cell and the cytosol fraction was also analyzed by western blotting.

**Microscopy and data analysis.** A PerkinElmer Ultraview Spinning Disk confocal microscope equipped with a Nikon Apochromat TIRF 100 × 1.49NA objective and a Hamamatasu C9100-23B EMCCD camera was used to acquire images and to perform live-cell imaging. Microscopic samples were prepared by sandwiching yeast cells between a 3% agarose pad and a coverslip[45]. The images were obtained by using Volocity software (PerkinElmer, Waltham, Massachusetts, USA) at room temperature. For stack images, 11 planes with 0.5 μm spacing were acquired. ImageJ 1.5 (imagej.nih.gov) and MetaMorph 7.7 (www.moleculardevices.com) softwares were used to analyze microscopy data. The plot graphs were created with KaleidaGraph 4.5 (www.synergy.com). Statistical analysis was performed with KaleidaGraph 4.5.

**Statistics and reproducibility.** Representative images from two independent experiments for Figs. 1a, 1d, 1e, 2a, 3a, 4e, 5e and Supplementary Figs. 1a, 1e, 2 (related to Fig. 2a), 3 (related to Fig. 4e) were shown. Western blotting assays were performed three times to test the expression of Emr1 from the promoters *emr1*, *ase1*, and *nmt41* (Fig. 2c). Western blotting assays were performed two times to test the expression of Mdm12-tdTomato (Fig. 4d) and Mmm1-tdTomato (Fig. 5d). Carbonate extraction and proteinase K assays shown in Fig. 2e, f were carried out two times.

**Reporting summary.** Further information on research design is available in the Nature Research Reporting Summary linked to this article.

## Data availability
All data generated or analyzed during this study are included in this published article, its supplementary information file, and the source data file. All protein and gene sequences of *Schizosacchromyces pombe* were obtained from the database Pombase (https://www.pombase.org). The sequence of Mco6 was obtained from the Saccharomyces Genome Database (SGD) (https://www.yeastgenome.org). The strains and plasmids used in this study are readily available upon request. All other information that support the findings of this paper are available upon reasonable request. Source data are provided with this paper.

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

## Acknowledgements

We thank Dr. Quanwen Jin for critical reading of the manuscript, Dr. Ying Huang for providing antibodies, and Dr. Chao Wang for the help of the biochemistry work. This work is supported by grants from the Strategic Priority Research Program of the Chinese Academy of Sciences (XDB19040101), National Natural Science Foundation of China (91754106, 31871350, 31671406, 31601095, and 31621002), National Key Research and Development Program of China (2017YFA0503600 and 2018YFC1004700), and China's 1000 Young Talents Recruitment Program.

## Author contributions

C.F. conceived and supervised the project. F.R., F.D., J.H., L.L., W.L., J.Y.C., W.W., and F. Z. performed experiments. W.W. and F.R. conducted biochemical work. F.R. analyzed data. F.R., W.W., and C.F. wrote the paper. All authors made comments.

## Competing interests

The authors declare no competing interests.
