## [Peer Review File · Nature Communications]

Reviewers' comments:

Reviewer #1 (Remarks to the Author):

The manuscript by Rasul et al describes an important new component of the ERMES complex – a tethering complex that supports the contact between the endoplasmic reticulum (ER) and mitochondria in yeast. To date no assembly protein has been found for this complex and this has remained an important open question in the field. In their manuscript the authors describe, Emr1 in pombe and suggest it as an ERMES assembly factor. The authors verified that Emr1 it is the functional homolog of Mco6 in cerevisiae.

While the paper addresses an important question, is very clearly and well written, and the results shown are very compelling and beautiful in my eyes ad demonstrate clearly that Emr1 interacts with the ERMES complex, co localizes with it and regulates foci number in cells – there is a major problem of over interpretation in my eyes. The fact that Mco6 is an assembly factor for ERMES is NOT shown in the paper. No data in the manuscript exists on ERMES assembly and only a proxy of the number of ERMES foci is used which could be also a read-out on the number of ER-Mitochondria contact sites in the cell suggesting Emr1 is a regulator of the number of contacts or a functional component of the complex. Also, in my eyes, the loss of mitochondrial shape may lead to reduced contact sites and this is not controlled for in other mutants that cause a similar collapse in shape.

Therefore, I can only support the publication of this important paper with the condition that the authors either directly show assembly of the complex is affected or alter the entire text of the manuscript to reflect what they actually show – which in my eyes is that Emr1 is a regulator of number of ERMES foci. This has to be a global change from title to text wording as their (over) interpretation of the results, not supported by the experiments themselves, goes throughout the manuscript.

Importantly – In my eyes when all such textual changes will be made to what the authors really see, the paper would still merit publication in Nature Communications as this finding is exciting in my eyes even without the over-interpretation.

Some examples below:

1. Introduction: The authors write that “the (ERMES) complex is responsible for creating ER-mitochondria contact sites” but this is not true. ER-Mitochondria contact sites can occur without presence of ERMES. Please rephrase.
2. In the section “Emr1 regulates proper assembly of the ERMES complex” the authors demonstrate a reduction in the number of ERMES foci (by two markers) and then state “Together, these results suggest that Emr1 plays a crucial role in regulating ERMES assembly”. I believe that this is not the case. At present their results only suggest that Emr1 plays some role in regulating ERMES foci number. This could be through ERMES assembly but also through many other routes. Please rephrase in title and in main text.
3. Also at the end of this section the authors write “Collectively, these data support the conclusion that Emr1 interacts with the ERMES complex and is required for the proper assembly of the ERMES

complex". This is also an overstatement – their data, until this point, although very compelling in my eyes that emr1 interacts with ERMES components, has not yet proven in any way that it affects assembly. At this point in the data, It could regulate it in many other ways or be a functional component (for example transferring something between the two organelles). Please rephrase to give accurate interpretation at each point in the manuscript and not over-interpret data presented this far.

4. In the next section "The C-terminus of Emr1 is required for directing proper assembly of the ERMES complex" the authors state again following an assay for number of foci that "confirming that the C terminus of Emr1 is required for proper assembly of the ERMES complex" – which is still not true. This assay only confirms that the C terminus of Emr1 is required for determining ERMES foci number. Please rephrase. In the same section you repeat the over interpretation many many times ("was able to rescue defective ERMES assembly" "if the C-terminus alone is enough for directing the assembly of the ERMES complex" "These results suggest that the Emr1 C-terminus alone is not sufficient for directing assembly of the ERMES complex." "is not sufficient for efficient assembly of the ERMES complex" "is required for directing efficient assembly of the ERMES complex" Please change all of these to a terminology that describes the effect on the number of Mdm12 foci, which is what you show.

5. The title "Artificial tethering of mitochondria to the ER restores normal mitochondrial morphology but not ERMES assembly in emr1Δ cell" is again an over interpretation. And in the content of this paragraph the same applies.

6. In the discussion. I totally agree with the phrasing of the authors "We provide strong evidence to support that Emr1 serves as a crucial regulatory subunit of the ERMES complex." But disagree that "Emr1 is the first reported crucial regulatory protein required for ERMES formation" because they do not show in my eyes an effect on ERMES formation – only on ERMES foci number – which could be an indirect effect of many other functions. This the authors could discuss and raise (post translational modification? A transfer function? A positioning role?)

7. In figure 3D the authors show that Emr1 does not colocalize with all ERMES foci and this should be clearly stated and discussed.

Small suggestion to authors that is not related to review of the manuscript – When the paper is accepted it would be important for the authors to convey this information also to the cerevisiae community. Potentially through SGD and potentially through changing the cerevisiae homolog name to Emr1 as well.

Reviewer #2 (Remarks to the Author):

In this manuscript, authors discover a novel regulator of ERMES (endoplasmic reticulum mitochondria encounter structure) complex, named Emr1. They suggest that Emr1 is integrated into the mitochondrial outer membrane, interacts with ERMES complex, and consequently regulates mitochondrial morphology. Since abnormal mitochondrial morphology of emr1Δ is restored by expression of artificial mitochondria-ER tether, Emr1 seems to be required for mitochondria-ER tethering function of ERMES complex.

The findings are interesting and well analyzed. This work has few concerns that should be addressed.

Major points

1. Topology of Emr1: In figure 2d, authors performed a proteinase K digestion assay to clarify Emr1 topology. GFP-Emr1 is shown to be resistant to the protease, and authors concluded that Emr1 is transmembrane protein on mitochondrial outer membrane. If this is the case, GFP-Emr1 in proteinase K-treated mitochondria should be 17 aa shorter, but the difference is too small to discriminate because of the size of GFP (238 aa). Therefore, the results also indicate other possibilities including GFP-Emr1 localizes in the mitochondrial matrix as Mti3 (Luo et al., 2019, FEBS J), in the intermembrane space, or on the inner membrane. The authors should distinguish from these possibilities. To verify whether Emr1 is a transmembrane protein, the carbonate extraction should be done.
2. In figure 3e, authors performed co-immunoprecipitation assay but in input there are too many unspecific bands detected by anti-GFP to evaluate the amount of prey. In addition, in condition (c), several bands are detected by anti-tdTomato. Since the results are one of the most important results in this manuscript, the authors should show more clear results in order to convince readers that Emr1 interact with ERMES specifically.
3. Authors indicate that Emr1 regulates mitochondria-ER contact sites through the assembly of ERMES complex (figure 3). The authors should elucidate whether *emr1* Δ affects the function of MERCs, such as lipid transport/synthesis.
4. In figure 4, authors test which of Emr1 mutants can rescue the Mdm12 puncta reduction in *emr1* Δ and revealed that Emr1- Δ N (=TM+C) recovered the phenotype but Emr1- Δ C (=TM+N), GFP-Emr1(C), and ER-GFP-Emr1(C) did not recover it completely. From these results, the authors concluded the cytosolic/C-terminal region is required for the ERMES complex assembly only when it localizes on mitochondria. However, ER-GFP-Emr1(C) shows a little recovery, and GFP moiety (238 aa) might affect Emr1(C) (17 aa) function because of its size. The authors should test with smaller tags. Furthermore, to reach the authors conclusion, mitochondria-targeted Emr1(C) should be tested.

Minor points

1. In figure S1A, 2e, 4b, 5d, authors should analyze mitochondrial morphology with the classification as demonstrated in figure 1b.
2. Colocalization analysis: Authors show colocalization with Z projection images, such as Emr1 and Cox4 in figure 2a or Emr1 and Mdm12 in figure 3d. The authors should show Z stack images without projection, to verify their colocalization.
3. In figure 3f, authors should show the blotting by anti-GST antibody to allow the evaluation of the bait's amount, in addition to Coomassie blue staining.
4. To test statistical significance among more than 2 conditions (Figure 4c, 4e, 5d and S4B), authors should not use Student's t-test, to avoid type I error (e.g. ANOVA is recommended).
5. Authors should show how much *emr1* (fission yeast) sequence is conserved with *mco6* (budding yeast), and discuss the reason why *mco6* show partial recovery of *mdm12* focus number (Figure S4B).
6. Authors should specify whether thiamine treated to GFP-chimera expressing cells shown in figure 5d and e, and explain the difference of GFP-Chimera localization between figure 5b and d.

Responses to reviewers' comments

We would like to thank both reviewers for providing the helpful suggestions. As you will see from the revised manuscript, we have made a strong effort to address all concerns raised. Specifically, we have added in the revised manuscript 11 new figures (Figs. 2e, 2f, 2h, 3e, 3f, 5b, 5d, 5f, 5g, 5h, and 5i), and 9 new supplementary figures (Supplementary Figs. 1c, 2a, 3e, 4a, 4b, 4e, 5a, 5b and 5c). In addition, we have modified several figures as suggested. The changed text is highlighted in blue in the revised manuscript and the point-by-point responses to the raised concerns are also highlighted in blue below.

Reviewers' comments:

Reviewer #1 (Remarks to the Author):

The manuscript by Rasul et al describes an important new component of the ERMES complex – a tethering complex that supports the contact between the endoplasmic reticulum (ER) and mitochondria in yeast. To date no assembly protein has been found for this complex and this has remained an important open question in the field. In their manuscript the authors describe, Emr1 in pombe and suggest it as an ERMES assembly factor. The authors verified that Emr1 it is the functional homolog of Mco6 in cerevisiae.

While the paper addresses an important question, is very clearly and well written, and the results shown are very compelling and beautiful in my eyes ad demonstrate clearly that Emr1 interacts with the ERMES complex, co localizes with it and regulates foci number in cells – there is a major problem of over interpretation in my eyes. The fact that Mco6 is an assembly factor for ERMES is NOT shown in the paper. No data in the manuscript exists on ERMES assembly and only a proxy of the number of ERMES foci is used which could be also a read-out on the number of ER-Mitochondria contact sites in the cell suggesting Emr1 is a regulator of the number of contacts or a functional component of the complex. Also, in my eyes, the loss of mitochondrial shape may lead to reduced contact sites and this is not controlled for in other mutants that cause a similar collapse in shape.

Therefore, I can only support the publication of this important paper with the condition that the authors either directly show assembly of the complex is affected or alter the entire text of the manuscript to reflect what they actually show – which in my eyes is that Emr1 is a regulator of number of ERMES foci. This has to be a global change from title to text wording as their (over) interpretation of the results, not supported by the experiments themselves, goes throughout the manuscript.

Importantly – In my eyes when all such textual changes will be made to what the authors really see, the paper would still merit publication in Nature Communications as this finding is exciting in my eyes even without the over-interpretation.

We thank the reviewer for the positive comments. In the revised manuscript, we followed the reviewer's comments to rephrase the sentences of concern in the entire text to reflect what the data show. Specifically, in the revised manuscript, we demonstrate that Emr1 plays a critical role in regulating the number of ERMES foci within the cell.

Some examples below:

1. Introduction: The authors write that “the (ERMES) complex is responsible for creating ER-mitochondria contact sites” but this is not true. ER-Mitochondria contact sites can occur without presence of ERMES. Please rephrase.

The original statement has been rephrased:

“In yeasts, the endoplasmic reticulum (ER)-mitochondria encounter structure (ERMES) complex **plays an important role in mediating** the formation of ER-mitochondria contact sites” (Introduction, the 2nd paragraph on Page 3)

2. In the section “Emr1 regulates proper assembly of the ERMES complex” the authors demonstrate a reduction in the number of ERMES foci (by two markers) and then state “Together, these results suggest that Emr1 plays a crucial role in regulating ERMES assembly”. I believe that this is not the case. At present their results only suggest that Emr1 plays some role in regulating ERMES foci number. This could be through ERMES assembly but also through many other routes. Please rephrase in title and in main text.

We agree with the reviewer that Emr1 plays an important role in regulating the number of ERMES foci. The title and the main text in this section have been changed accordingly:

- 1) Title: “Emr1 regulates **the number of ERMES foci**”.
- 2) Main text: a) “This prompted us to ask whether Emr1 is involved in regulating ERMES **functions**”, b) “Together, these results suggest that Emr1 plays a crucial role in regulating **the number of ERMES foci within the cell**”.

3. Also at the end of this section the authors write “Collectively, these data support the conclusion that Emr1 interacts with the ERMES complex and is required for the proper assembly of the ERMES complex”. This is also an overstatement – their data, until this point, although very compelling in my eyes that emr1 interacts with ERMES components, has not yet proven in any way that it affects assembly. At this point in the data, It could regulate it in many other ways or be a functional component (for example transferring something between the two organelles). Please rephrase to give accurate interpretation at each point in the manuscript and not over-interpret data presented this far.

We have followed the suggestion to rephrase the statement:

“Collectively, these data support the conclusion that Emr1 interacts with the ERMES complex and regulates the number of ERMES foci”.

4. In the next section “The C-terminus of Emr1 is required for directing proper assembly of the ERMES complex” the authors state again following an assay for number of foci that “confirming that the C terminus of Emr1 is required for proper assembly of the ERMES complex” – which is still not true. This assay only confirms that the C terminus of Emr1 is required for determining ERMES foci number. Please rephrase. In the same section you repeat the over interpretation many many times (“was able to rescue defective ERMES assembly” “if the C-terminus alone is enough for directing the assembly of the ERMES complex” “These results suggest that the Emr1 C-terminus alone is not sufficient for directing assembly of the ERMES complex.” “is not sufficient for efficient assembly of the ERMES complex” “is required for directing efficient assembly of the ERMES complex” Please change all of these to a terminology that describes the effect on the number of Mdm12 foci, which is what you show.

We have followed the reviewer’s suggestion to rephrase the sentences of concern and the changes are highlighted in blue in the main text.

5. The title “Artificial tethering of mitochondria to the ER restores normal mitochondrial morphology but not ERMES assembly in emr1 Δ cell” is again an over interpretation. And in the content of this paragraph the same applies.

The sentences of concern have been rephrased and have been highlighted in blue in the main text.

6. In the discussion. I totally agree with the phrasing of the authors “We provide strong evidence to support that Emr1 serves as a crucial regulatory subunit of the ERMES complex.” But disagree that “Emr1 is the first reported crucial regulatory protein required for ERMES formation” because they do not show in my eyes an effect on ERMES formation – only on ERMES foci number – which could be an indirect effect of many other functions. This the authors could discuss and raise (post translational modification? A transfer function? A positioning role?)

The sentence of concern have been rephrased as follows:

“Therefore, Emr1 is a crucial regulatory protein required for proper functions of the ERMES complex”

In addition, we have discussed in the paragraph before the last one in the discussion how Emr1 could be involved in regulating the number of ERMES foci and/or the assembly of the ERMES complex.

7. In figure 3D the authors show that Emr1 does not colocalize with all ERMES foci and this

should be clearly stated and discussed.

We have added the new statements below to the result section (page 6).

“It appeared that GFP-Emr1 did not completely colocalized with Mdm12 and some GFP-Emr1 signals were present on mitochondria (Fig. 3d). This is likely due to the higher expression levels of GFP-Emr1 from the *ase1* promoter than from its own promoter (Fig. 2a,c)”

Small suggestion to authors that is not related to review of the manuscript - When the paper is accepted it would be important for the authors to convey this information also to the cerevisiae community. Potentially through SGD and potentially through changing the cerevisiae homolog name to Emr1 as well.

We thank the reviewer for the reminder. After the manuscript is accepted for publication, we will write to SGD to request a change for the name of Mco6 to Emr1 and to annotate Mco6/Emr1 in SGD.

Reviewer #2 (Remarks to the Author):

In this manuscript, authors discover a novel regulator of ERMES (endoplasmic reticulum mitochondria encounter structure) complex, named Emr1. They suggest that Emr1 is integrated into the mitochondrial outer membrane, interacts with ERMES complex, and consequently regulates mitochondrial morphology. Since abnormal mitochondrial morphology of *emr1Δ* is restored by expression of artificial mitochondria-ER tether, Emr1 seems to be required for mitochondria-ER tethering function of ERMES complex.

The findings are interesting and well analyzed. This work has few concerns that should be addressed.

We would like to thank the reviewer for the kind support.

Major points

1. Topology of Emr1: In figure 2d, authors performed a proteinase K digestion assay to clarify Emr1 topology. GFP-Emr1 is shown to be resistant to the protease, and authors concluded that Emr1 is transmembrane protein on mitochondrial outer membrane. If this is the case, GFP-Emr1 in proteinase K-treated mitochondria should be 17 aa shorter, but the difference is too small to discriminate because of the size of GFP (238 aa). Therefore, the results also indicate other possibilities including GFP-Emr1 localizes in the mitochondrial matrix as Mti3 (Luo et al., 2019, FEBS J), in the intermembrane space, or on the inner membrane. The authors should distinguish from these possibilities. To verify whether Emr1 is a transmembrane protein, the carbonate extraction should be done.

We followed the suggestions to analyze Emr1 topology and the mitochondrial localization of Emr1 both by carbonate extraction and by proteinase K digestion assays (see new Figs. 2e and 2f in the revised manuscript).

- 1) Carbonate extraction. Isolated mitochondria were extracted by carbonate. As shown in new Fig. 2e, similar to Yta4-13myc, GFP-Emr1 was present in the insoluble fraction of the carbonate extraction (membrane fraction) whereas the mitochondrial matrix protein Mti2 was present in the soluble fraction. This result unambiguously supports the claim that Emr1 is an integral mitochondrial membrane protein.

- 2) To further distinguish the membrane of mitochondria where GFP-Emr1 is localized, we performed rigorous experiments of proteinase K digestion. As shown in new Fig. 2f, isolated mitochondria were incubated in isotonic or hypotonic buffers with or without proteinase K. In addition, isolated mitochondria were incubated in a hypotonic buffer containing Triton X-100 to extract membrane proteins and were treated with or without proteinase K. In the isotonic buffer condition, the mitochondrial outer membrane remained intact, and therefore a band of full-length GFP-Emr1 was detected above 34 kDa and a smaller band of GFP-Emr1, presumably the digested GFP-Emr1 lacking the cytoplasmic region, was detected below 34 kDa after proteinase K treatment. This result suggests that GFP-Emr1 likely resides on the mitochondrial outer membrane.

Consistently, in the hypotonic buffer condition, the mitochondrial outer membrane ruptured, leading to the exposure of the GFP tag of GFP-Emr1 to proteinase K. As a result, GFP-Emr1 was digested more efficiently (the intensity of the bands above and below 34 kDa reduced significantly). This result was similar to the one obtained in the hypotonic plus Triton X-100 condition. In these series of experiments, Yta4-13Myc, the tag Myc and a large portion of the protein are outside of mitochondria, and the mitochondrial matrix protein Mti2 were used as controls, and the results were as expected, confirming the success of the experiments.

Finally, we noticed that the GFP-Emr1 band below 34 kDa also appeared in the whole cell lysate and the cytosol fraction from mitochondrial isolation. We speculated that the cytoplasmic region of Emr1 may be prone to degradation during the biochemical purification processes.

Overall, the results support the claims that Emr1 is an integral mitochondrial membrane protein and localizes to the mitochondrial outer membrane with its C-terminus exposed in the cytoplasm. The relevant statements have been added in the result section of the revised manuscript (Page 5).

2. In figure 3e, authors performed co-immunoprecipitation assay but in input there are too many unspecific bands detected by anti-GFP to evaluate the amount of prey. In addition, in condition (c), several bands are detected by anti-tdTomato. Since the results are one of the most important results in this manuscript, the authors should show more clear results in order to convince readers

that Emr1 interact with ERMES specifically.

We agreed with the reviewer that the co-IP result is one of the most important results. Therefore, we have repeated the co-IP experiments multiple times during the revision. The results consistently showed that GFP-Emr1 co-precipitated well with Mdm12, but less so with mmm1, as shown in new Fig. 3e.

3. Authors indicate that Emr1 regulates mitochondria-ER contact sites through the assembly of ERMES complex (figure 3). The authors should elucidate whether *emr1Δ* affects the function of MERCs, such as lipid transport/synthesis.

We followed the reviewer's suggestions to develop TLC (Thin layer chromatography) assays in the lab. TLC analysis of phospholipids (PS, PE, and CL) extracted from whole cells and isolated mitochondria showed that PE is significantly reduced in *emr1Δ* cells and that PS was slightly but significantly reduced in isolated mitochondria in *emr1Δ* cells. These results suggest that the absence of Emr1 indeed affects the function of MERCs, likely through regulating the function of ERMES as shown in Fig. 3.

PE appeared to be mainly synthesized in mitochondria by the PS decarboxylase Psd1, using PS transported from the ER through MERCs^{1,2}. Fission yeast cells appear to have at least two other pathways to produce PE, though not well characterized^{1,2} (see diagram in supplementary Fig. 5c): 1) Kennedy pathway (Ept1 is one of the critical enzymes in the pathway), and 2) other cellular compartments except mitochondria by the PS decarboxylases Psd2 and Psd3. To strengthen our claim that Emr1 affects lipid transport between the ER and mitochondria, epistasis analysis was carried out. As shown in supplementary Fig. 5c, *psd1Δ* and *emr1Δ* did not have an additive effect on cell growth, whereas *emr1Δ* has an additive effect with *psd2Δ*, *psd3Δ*, or *ept1Δ* on cell growth. This epistasis analysis supports the idea that Emr1 and Psd1 work in the same route to produce PE but in parallel with Psd2/3/Ept1. Hence, our new data unambiguously support the hypothesis that *emr1Δ* affects the function of MERCs, leading to abnormal lipid transport/synthesis. These new results and relevant statements have been added in the result section of the revised manuscript (page 7 above Discussion).

4. In figure 4, authors test which of Emr1 mutants can rescue the Mdm12 puncta reduction in *emr1Δ* and revealed that Emr1-ΔN (=TM+C) recovered the phenotype but Emr1-ΔC (=TM+N), GFP-Emr1(C), and ER-GFP-Emr1(C) did not recover it completely. From these results, the authors concluded the cytosolic/C-terminal region is required for the ERMES complex assembly only when it localizes on mitochondria. However, ER-GFP-Emr1(C) shows a little recovery, and GFP moiety (238 aa) might affect Emr1(C) (17 aa) function because of its size. The authors should test with smaller tags. Furthermore, to reach the authors conclusion, mitochondria-targeted Emr1(C) should be tested.

We have followed the reviewer's suggestions to characterize the Emr1 truncated mutants further.

1) Using smaller tags to bridge the ER-targeting module and Emr1(C). We took two new

designs. As illustrated in new supplementary Fig. 4a, the ER-targeting module was directly fused to Emr1(C) to create a new chimera (i.e. ER-Emr1(C)) and the ER-targeting module was linked to Emr1(C) by a smaller tag, i.e. 13Myc (~16 kDa) to generate ER-13Myc-Emr1(C). Similar to ER-GFP-Emr1(C) (see Fig. 4d,e), ER-Emr1(C) and ER-13Myc-Emr1(C) only partially restored the number of Mdm12 foci in *emr1Δ* cells (see new supplementary Fig. 4a). These results suggest that the linking sequences may not affect the function of Emr1(C).

- 2) Using a mitochondria-targeting module to target Emr1(C) to mitochondria. We fused the transmembrane domain of Yta4, as shown in Fig. 2d, to the N-terminus of GFP-Emr1(C) (named Yta4(TM)-GFP-Emr1(C)). Surprisingly, although Yta4(TM)-GFP-Emr1(C) successfully localized to mitochondria, it also only partially restored the number of Mdm12 foci in *emr1Δ* cells (see new supplementary Fig. 4b). This result, together with the ones in Fig. 4c,e, suggests that the cytoplasmic region and the transmembrane domain are required for Emr1 to be fully functional. Therefore, in the revised manuscript, we have changed our original claim to the statement below (page 6):

“Hence, we concluded that the cytoplasmic region of Emr1 plays an important role in regulating the number of ERMES foci but both the cytoplasmic region and the transmembrane domain are required for Emr1 to be fully functional.”

Minor points

1. In figure S1A, 2e, 4b, 5d, authors should analyze mitochondrial morphology with the classification as demonstrated in figure 1b.

We have done quantification for Fig. S1A (see result in new supplementary Fig. 1c), 2e (now 2g; see result in new Fig. 2h), 4b (see result in new supplementary Fig. 2a) and 5d (see result in new supplementary Fig. 5b).

2. Colocalization analysis: Authors show colocalization with Z projection images, such as Emr1 and Cox4 in figure 2a or Emr1 and Mdm12 in figure 3d. The authors should show Z stack images without projection, to verify their colocalization.

The Z stack images without projection were shown in new supplementary Fig. 2a (related to Fig. 2a) and new supplementary Fig. 3e (related to Fig. 3d).

3. In figure 3f, authors should show the blotting by anti-GST antibody to allow the evaluation of the bait' s amount, in addition to Coomassie blue staining.

We have followed the suggestion to perform the GST pull-down experiment again and probed the membrane with an anti-GST antibody. As shown in new Fig. 3f, Mdm12-GST and Mdm34, but not GST alone, physically interacted with His-GFP-Emr1.

4. To test statistical significance among more than 2 conditions (Figure 4c, 4e, 5d and S4B),

authors should not use Student's t-test, to avoid type I error (e.g. ANOVA is recommended).

We have followed the suggestion to perform ANOVA analysis for the indicated data.

5. Authors should show how much *emr1* (fission yeast) sequence is conserved with *mco6* (budding yeast), and discuss the reason why *mco6* show partial recovery of *mdm12* focus number (Figure S4B).

We have added new supplementary Fig. 4e to show alignment between *Mco6* and *Emr1*. As shown in supplementary Fig. 4e, the amino acid identity between the two proteins is ~30% and *Mco6* has a rather short C-terminus. These may be the reasons why *Mco6* only partially rescued the number of *Mdm12* foci. The statements have been added to the result section in the revised manuscript (page 6).

6. Authors should specify whether thiamine treated to GFP-chimera expressing cells shown in figure 5d and e, and explain the difference of GFP-Chimera localization between figure 5b and d.

To address the concern, we performed imaging experiments in parallel for the cells shown in Fig. 5b and 5d. In the revised manuscript, we have followed the suggestion to clearly mark in figures and to state in figure legends whether thiamine was used in the culture. In addition, for fair comparison, in new Fig. 5b and 5d, both maximum projection images of the GFP channel and the middle-plane images from the respective Z-stack were shown. The localization of the GFP tether chimera expressed in the two different strains is similar, i.e. to the ER. In addition, the GFP tether chimera displayed dense aggregation signals in both types of cells but the aggregation signals were more obvious in the cells co-expressing *Cox4-RFP* (Fig. 5d). Nonetheless, expression of the tether did not rescue the decreased number of *Mdm12* foci in *emr1Δ* cells but largely restored mitochondrial morphology (Fig. 5c, e and supplementary Fig. 5b).

References

1. Holic, R., Pokorna, L. & Griac, P. Metabolism of phospholipids in the yeast *Schizosaccharomyces pombe*. *Yeast* **37**, 73-92 (2020).
2. Flis, V.V. & Daum, G. Lipid transport between the endoplasmic reticulum and mitochondria. *Cold Spring Harb Perspect Biol* **5** (2013).

REVIEWERS' COMMENTS

Reviewer #1 (Remarks to the Author):

The authors have now fully answered all of my concerns and I support the publication of the manuscript in its current form.

Reviewer #2 (Remarks to the Author):

Authors convincingly addressed my concerns. This paper represents a valuable addition to the field.

Responses to reviewers' comments

We would like to thank the two reviewers again for their kind support. Now both reviewers support the publication of the work.

REVIEWERS' COMMENTS

Reviewer #1 (Remarks to the Author):

The authors have now fully answered all of my concerns and I support the publication of the manuscript in its current form.

Reviewer #2 (Remarks to the Author):

Authors convincingly addressed my concerns. This paper represents a valuable addition to the field.